# Landslide activation during deglaciation in a fjord-dominated landscape: observations from southern Alaska (1984-2022)

Jane Walden[1,2], Mylène Jacquemart[1,2], Bretwood Higman[3], Romain Hugonnet[1,2,4], Andrea Manconi[5,6], and Daniel Farinotti[1,2]

[1]Laboratory of Hydraulics, Hydrology and Glaciology (VAW), Swiss Federal Institute of Technology (ETH), Zurich, Switzerland
[2]Swiss Federal Institute for Forest, Snow and Landscape Research (WSL), bâtiment ALPOLE, Sion, Switzerland
[3]Ground Truth Trekking, Seldovia, AK, USA
[4]Department of Civil and Environmental Engineering, University of Washington, Seattle, WA, USA
[5]Institute for Snow and Avalanche Research (SLF), Swiss Federal Institute of Forest, Snow and Landscape Research (WSL), Davos, Switzerland
[6]Department of Earth and Planetary Sciences, Swiss Federal Institute of Technology (ETH) Zurich, Zurich, Switzerland

**Correspondence:** Jane Walden (walden@vaw.baug.ethz.ch)

**Abstract.** A consequence of the current global glacier mass loss is the destabilization of valley walls as the support provided by the glacier evolves and eventually vanishes, a process typically referred to as "debuttressing." In this work, we examined the evolution of eight large, active landslides in southern coastal Alaska, a region experiencing some of the fastest glacier mass loss worldwide. Additionally, many glaciers in this area are retreating out of glacially carved fjords, leaving landslides
in contact with deep water bodies that can substantially increase the reach of a catastrophic failure. We used automatic and manual feature tracking on optical imagery to derive slope movement from the 1980s to present and compared this with glacier terminus retreat and thinning, precipitation, and seismic energy, paying particular attention to landslides in contact with lake or ocean water. We found that the majority of landslides underwent a pulse of accelerated motion during the studied time period. In four cases, landslide movement coincided with the rapid retreat of a lake- or marine-terminating glacier past the instability. At
these sites and during these accelerations, the glacier retreat rates were up to 7 times higher than average, while the landslides reached velocities that were up to 9 times higher than their long-term average. Two sites showed no movement, though both landslides are known to be moving at velocities below the detection threshold of the methods employed here. At two other sites where the landslides are still in contact with the ice, above-average precipitation and increased glacier thinning were found to coincide with accelerated motion, though conclusive causal links could not be drawn and the effect of short-term precipitation
could not be ruled out. Our results suggest that landslides adjacent to lakes or fjords may be especially susceptible to sudden activation, which we propose is due to the particularly rapid retreat rates of water-terminating glaciers as well as mechanical and hydrological changes resulting from the replacement of ice with water at the landslide toe in relatively short timescales. By showing that glacier mass loss is associated with increased landslide movement across various settings in Alaska, we suggest that glacier-landslide interactions in coastal settings deserve special attention and further substantiate the need for establishing
broader and more systematic paraglacial hazard monitoring in a warming world.

# 1 Introduction

Anthropogenic climate change is causing rapid glacier thinning and retreat all over the world (IPCC, 2022). Glacier mass loss has a wide variety of impacts on the Earth system and human livelihoods, diminished water resources and sea level rise being among the most consequential (IPCC, 2019; Immerzeel et al., 2020). Glacier retreat, which removes support from adjacent valley walls in a process termed "glacier debuttressing" (Ballantyne, 2002), may lead to the destabilization or failure of weakened valley slopes. There is limited direct risk to humans from such slow-moving slopes, since these areas are too dynamic to support human infrastructure. However, catastrophic slope failure can have significant downstream impacts by damming rivers or abruptly increasing sediment input into proglacial water systems (Fan et al., 2020). Additionally, landslides can initiate highly mobile cascading processes which may threaten critical infrastructure (e.g., Van Wyk de Vries et al., 2022; Shugar, D.H. et al., 2021; Sharma et al., 2023).

Where catastrophic landslides enter water, such as proglacial lakes or fjords, they can generate tsunamis. Albeit comparatively rare, landslide-tsunami cascades can have far-reaching and destructive impacts, as is exemplified by several cases from the last decades. In 1958, for example, a landslide near the terminus of Lituya Glacier, Alaska, impacted Lituya Bay, generating a tsunami that ran up $530\,\text{m}$ on a nearby ridge and killed two people (Miller, 1960). Less than a decade later, an instability near Grewingk Glacier, Alaska, failed catastrophically into a proglacial lake and caused a tsunami with $60\,\text{m}$ runup (Wiles and Calkin, 1992; Lemaire et al., 2023a). In 2000, a large landslide entered the Vaigat Strait in western Greenland, generating a tsunami with a $28\,\text{m}$ runup in a town $20\,\text{km}$ away from the source (Dahl-Jensen et al., 2004). In 2015, a large landslide in Taan Fiord, Alaska, caused a tsunami with $193\,\text{m}$ runup (Higman et al., 2018). Two years later, in Karrat Isfjord, western Greenland, a large landslide failed catastrophically into a fjord, generating a tsunami with $90\,\text{m}$ runup near to the failure site, and the tsunami inundated a village $32\,\text{km}$ away with $1$ to $1.5\,\text{m}$ waves, killing four people (Strzelecki and Jaskólski, 2020). More recently, in 2023, a large landslide impacted a fjord in Dickson Fjord, eastern Greenland, causing a $200\,\text{m}$ runup followed by a seiche which generated a seismic signal lasting for 9 days (Svennevig et al., 2024). Despite the destructive potential of these events, few studies have investigated landslide evolution near deep fjords with a specific focus on the glacier evolution.

Much remains unknown about how glacial erosion and debuttressing interact with other factors to precondition or trigger slope failures. In fact, there has been some debate about whether glacier debuttressing can cause slope failure due to the viscous nature of ice at low strain rates (McColl et al., 2010; McColl and Davies, 2013; Storni et al., 2020). Others suggest that debuttressing can increase shear stress and act in combination with other processes such as rainfall to promote slope movement (Le Roux et al., 2009). At Taan Fiord, above-average rainfall and an earthquake were identified as potential triggering factors, though the authors note that the glacier retreated over $17\,\text{km}$ in 50 years and thinned by $400\,\text{m}$ from 1961–1991 (Higman et al., 2018). At Grewingk Lake, a specific trigger could not be identified, though it is thought that the slope was weakened by a large earthquake in 1964, a month of intense precipitation, and multiple cycles of glacier retreat (Wiles and Calkin, 1992; Lemaire et al., 2023a). Precipitation can cause groundwater fluctuations which impact landslide stability (e.g., Handwerger et al., 2019a; Iverson, 2000) and seismic shaking has been shown to weaken slopes (Keefer, 1984; Lacroix et al., 2014), resulting in varying behavior during and after earthquakes (Kohler and Puzrin, 2023). In addition, litho-structural characteristics (Kuhn et al., 2023;

Stead and Wolter, 2015), rock mass properties (Wang et al., 2021; Gischig et al., 2016; Hugentobler et al., 2022), and changing lake water levels (Hendron and Patton, 1987; Wang et al., 2008) are among the mechanisms which drive landslide motion. All of these processes—as well as combinations of them—may be relevant to the sites studied here.

Connections between glacier and landslide changes have been documented at several sites around the world. At the Barry Arm landslide in southern Alaska, dramatic landslide acceleration was correlated with rapid glacier retreat (Dai et al., 2020). Interestingly, the kinematic response of the slope to deglaciation has changed throughout time (Schaefer et al., 2023). In a more alpine setting, the landslide at Tungnakvíslarjökull in Iceland sped up after the glacier mass loss increased, and glacier debuttressing was determined to be the main cause of slope acceleration (Lacroix et al., 2022). Studies of the Moosfluh landslide in Switzerland showed that landslide deformation can be related to debuttressing, with landslides reacting rapidly to glacier changes upon crossing a threshold of ice loss (Kos et al., 2016). Others found that the glacier controls the landslide velocity but has little effect on its stability (Storni et al., 2020), and Glueer et al. (2020) suggested that altered groundwater conditions may lead to enhanced slope instability. Numerical models show that thermo- and hydro- mechanical stresses from repeated glacier cycles weaken rock (Grämiger et al., 2018, 2020), and most rock damage occurs upon first deglaciation (Grämiger et al., 2017).

These examples demonstrate a connection between ice loss and slope stability and bring attention to a hazard that is not yet well understood. Alaska is a hotspot for glacier mass loss, making up $\sim 25\,\%$ of global glacier mass loss between 2000 and 2020 (Hugonnet et al., 2021), despite containing only around 12% of the world's glacier volume (Farinotti et al., 2019). Alaska has also come to light as a region with a precarious combination of paraglacial landslides and rapidly retreating tidewater or lake-terminating glaciers (Schaefer et al., 2024). (We use "paraglacial" to define non-glacial processes impacted by glaciation (Church and Ryder, 1972)). The increased awareness of this hazard spurred the creation of an Alaskan landslide inventory (Higman, 2022; Higman et al., 2023), which documents instabilities, relict landslides, and mass movement deposits throughout the state. Given that the high rates of glacier mass loss are projected to continue during the 21[st] century, it is of interest to determine how the risk posed by paraglacial landslides and potential tsunamis will change in the future.

To understand how glacier thinning and retreat control landslide activation and mobility, we take a synoptic view of eight large paraglacial landslides in southern Alaska (Sect. 2) and analyze their evolution using satellite imagery from 1984 until present, surface velocity changes from 1984–2022, and elevation changes from the mid-1900s until 2020. By doing so, we provide the first study comparing detailed glacier evolution—including both thinning and terminus retreat—with landslide movement in southern Alaska. Specifically, we ask whether the (re-)activation of paraglacial landslides can be explained by glacier retreat. In order to evaluate the relevance of the glacier changes, we also consider meteorological and seismic influences (Sect. 3). We combine these data to assess whether and how glacier changes can be responsible for slope destabilization (Sect. 4) and discuss these in the context of the possible physical mechanisms behind the slope instabilities (Sect. 5).

## 2  Study Area

Southern Alaska is heavily glaciated, hosting some of the largest glaciers in the world (Windnagel et al., 2023). Due to its proximity to the ocean, it has a maritime climate, leading to high annual precipitation and in turn, extensive glacier coverage

(Shulski and Wendler, 2007). Over the eight study sites (Fig. 1) and the period 1979–2022, average annual precipitation ranged from 2670 to 4030 mm while mean annual air temperatures varied between −4 and 3 °C (Hersbach et al., 2018). The large precipitation amounts result in a thick winter snowpack which, combined with relatively mild temperatures, make extended permafrost coverage unlikely – less than a 1% probability of occurrence at our sites according to Obu et al. (2018) (Fig. 1).

Alaska is a very seismically active region and experiences some of the highest uplift rates worldwide (Larsen et al., 2005). Glacier mass loss since the Little Ice Age and the location on a subduction zone result in uplift rates as high as $10 \, \mathrm{mm \, yr^{-1}}$ on the Kenai Peninsula (Cohen and Freymueller, 2001) and up to $32 \, \mathrm{mm \, yr^{-1}}$ in southeast Alaska (Larsen et al., 2005). The collision of the Pacific and North American plates has formed several major faults throughout the state. The Castle Mountain, Border Ranges, and Fairweather faults are relevant for this work (Mériaux et al., 2009). Alaska has had four of the twenty largest earthquakes recorded globally since 1900 (Earthquake Hazards Program, 2019), and an earthquake of magnitude eight or larger every 13 years since 1900 (Alaska Seismic Hazards Safety Commission, 2012). The state also sees many moderate earthquakes (magnitude 4–5): on average, 300 per year since 1900 (Alaska Seismic Hazards Safety Commission, 2012).

From the Higman (2022) inventory, we selected eight instabilities which are distributed throughout southern coastal Alaska from the Kenai Peninsula to Glacier Bay and have all been in contact with a glacier at some stage since the 1980s. The selection was relatively arbitrary, focusing on sites that stood out as worth investigating from early versions of the inventory. At four sites—Ellsworth, Portage, Columbia, and Alsek—the terminus has not yet retreated past the landslide area. At the other four sites—Barry, Yale, Tyndall, and Grand Plateau—the glacier has retreated past most or all of the landslide area, and these instabilities now border lakes or fjords. The sites have thus been impacted by glacier mass loss to varying degrees and in different ways, and therefore showcase a range of possible settings. All study sites are large landslides in sedimentary or metamorphic rock. Further description, including why each site was of particular interest, is found in Appendix A, and an overview of the sites can be seen in Figures 1 and 2. Note that we use the terms "landslide" and "instability" interchangeably throughout this publication.

## 3  Data & Methods

In order to understand how glacier thinning and retreat may control the motion of the investigated landslides, we analyzed slope deformation in response to glacier and environmental changes over the past 40 years. Specifically, we characterized their deformation from a combination of automatic and manual feature tracking on satellite optical imagery, and combined this with several parameters describing glacier change. We determined glacier retreat rates by mapping yearly terminus positions and quantified glacier thinning rates from digital elevation model (DEM) differencing. From these data, we evaluated whether any acceleration of the paraglacial landslide coincided with changes in glacier behavior. Finally, to gauge the importance of the glacier-related controls, we evaluated time series of precipitation and seismicity as possible co-drivers of landslide deformation. The details of the individual datasets and methods applied are presented in the following sections.

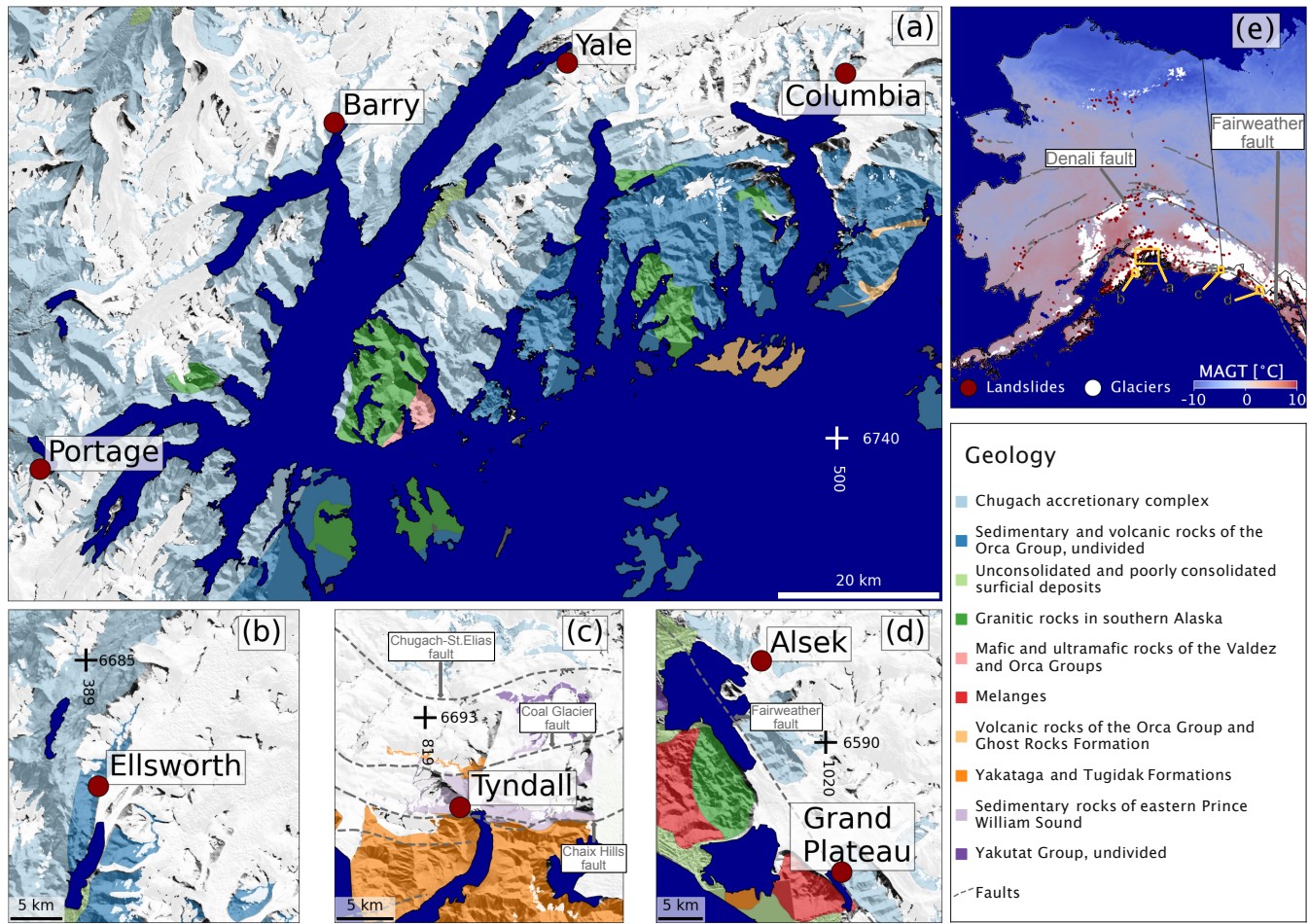

**Figure 1.** Overview of the study sites. Panels A-D (see panel E for location within Alaska) show the geology of the region (Wilson et al., 2015), with the legend at bottom right. The study sites are labeled with dark red points. Panel E shows the mean annual ground temperature (MAGT) (Obu et al., 2018), along with sites from Higman (2022) (red dots), glacierized area from RGI Consortium (2017) (white area), and faults from U.S. Geological Survey and Alaska Department of Natural Resources (2024) (gray dashed lines). Ocean area (dark blue) is from © OpenStreetMap contributors 2024. Distributed under the Open Data Commons Open Database License (ODbL) v1.0. A 2000s digital elevation model is used as a background layer (Hugonnet et al., 2021). A larger version of panel E is in the Supplementary Materials.

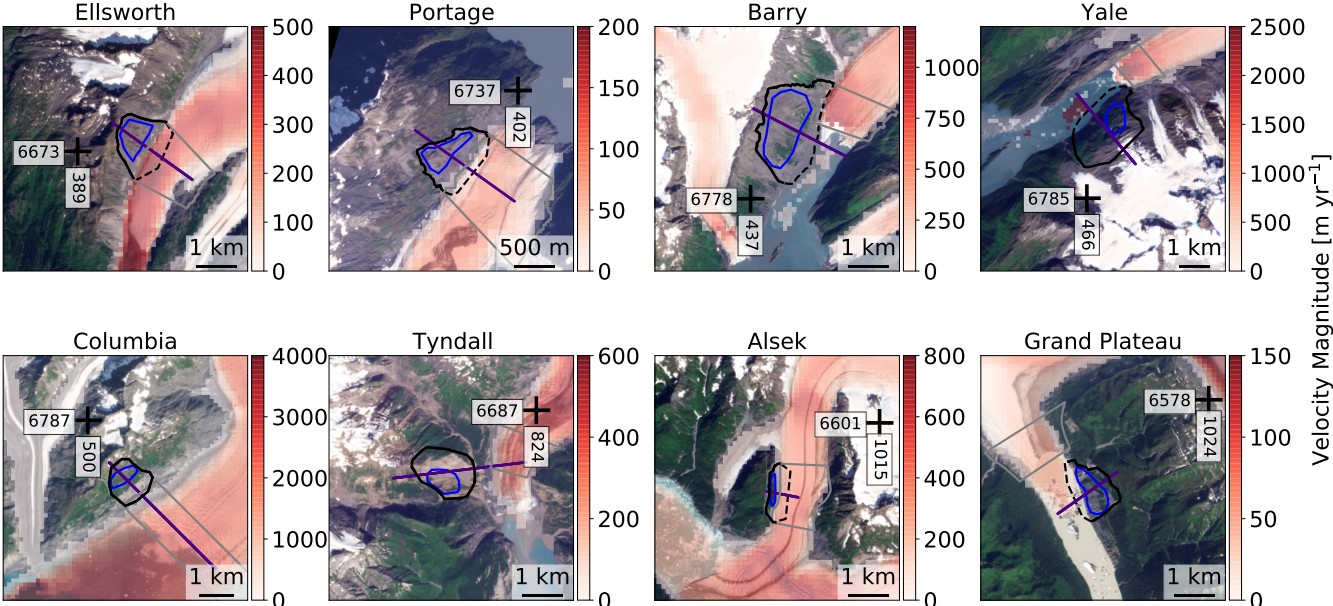

**Figure 2.** Satellite imagery from August 2023 over the eight sites (images © 2023 Planet Labs PBC). Instability outlines are shown in black, where dashed lines are the estimated subglacial or submarine instability extents. In the text, we refer to these outlines as "instability polygons" (see App. D for a description). The "active areas" (described in Sect. 3.1) are shown in blue, and the cross section (described in Sect. 3.2 and App. C) is plotted in purple. The instability-adjacent-glacier (IAG) polygon (see Sect. 3.2 and App. C) is plotted as a gray outline. Red shading represents the glacier speed from ITS-LIVE (Gardner et al., 2023), note that scales differ by more than one order of magnitude. Coordinates crosses refer to the UTM 6 projected coordinate system.

## 3.1 Landslide displacements

We retrieved landslide velocities from the ITS-LIVE annual mosaics (Gardner et al., 2023). The dataset was developed to map glacier flow but extends beyond the glacier boundaries, allowing for applications on nearby slopes. ITS-LIVE data is generated from Landsat 4-8 image pairs and processed using the autonomous Repeat Image Feature Tracking algorithm (auto-RIFT; (Gardner et al., 2018)). Between 1984 and 2022, the following relevant parameters are available each year at $120\,\text{m}$-resolution: velocity magnitude, velocity components in the x- and y- directions, and an estimate for the uncertainty in the velocity. Because

its primary use is on glaciers, the ITS-LIVE processing parameters are optimized for velocities that are much higher than for typical landslide motion. Additionally, the relatively coarse resolution of the dataset means that small displacements cannot be detected accurately. ITS-LIVE is thus expected to perform better for larger and fast-moving landslides, where surface features and resulting displacements are larger. Small changes or displacements of landslide are by no means irrelevant, but the resolution seems adequate for investigating the slope-wide responses of the landslides to glacier changes.

To retrieve landslide velocities from the ITS-LIVE data, we extracted the velocity and uncertainty over the "active area" of the instability to create an annual time series. The active area was delineated by selecting velocity pixels which fell within the instability polygon (see Fig. 2 for definition) and which showed movement in the 1984–2022 ITS-LIVE velocity magnitude data (Fig. 3). In some areas, the ITS-LIVE data appears to show "leakage" of the glacier signal to glacier-adjacent pixels, which may result from i) uncertainties in the glacier margins, ii) temporal inconsistencies between the glacier margins and the ITS-LIVE data, or iii) a large window size used during image cross-correlation. This leaked signal typically extends 1 to 2 pixels (i.e. up to $240\,\mathrm{m}$) away from the glacier and thus affects a small portion of the landslides' extents. When defining the active area, we excluded glacier pixels inside the RGI Consortium (2017) outlines, as well as high-velocity pixels moving parallel to the glacier, and thus do not expect this leakage to impact our results. At sites where no motion was shown by ITS-LIVE, we selected pixels over the area where most deformation is seen from satellite images.

Since we focus on landslide evolution over several decades—going back to the start of the satellite era—we leverage yearly and multi-year velocities to characterize landslide changes. The usage of annual data may result in the loss or smoothing of the signal, especially during times of rapid movement, and it does not allow us to see seasonal effects. However, this temporal resolution allows for characterization of long-term trends and interannual changes. Prior to 2000, the ITS-LIVE data has high uncertainties due to the availability of only lower resolution satellite data during this time. For this reason, we employ manual feature tracking over 5-year time steps prior to 2000, and then use the ITS-LIVE data supplemented by the manual feature tracking between 2000 and 2022.

We selected cloud- and snow-free summer Landsat images on average every 5 to 6 years (some images were as little as 3 years or as much as 12 years apart) and manually mapped displacements indicating coherent motion across the slope. We found this image spacing to provide large enough displacements to confidently map the changes (around 1 pixel or $30\,\mathrm{m}$) and highlight the slope's temporal evolution. We classified movement as "coherent" or "slope-wide" if large vegetated patches within the instability polygon synchronously moved downward (changes of isolated vegetation patches did not classify as movement). The "activation period" was defined manually to be the timing of an initial pulse of significant, slope-wide deformation during our study period (Sect. 5.2). For slopes that experienced catastrophic failure, no feature tracking was done during the time period containing the failure because the fundamental changes of the sliding mass make the pre- and post-failure slopes incomparable. For each landslide feature indicating slope-wide movement (e.g. crack opening, displacement of vegetation patches), we took around three to five measurements of the distance moved between two images. The median of these measurements, divided by the number of years between the images, gives the slope speed for that period.

### 3.2 Glacier elevation changes

To quantify glacier thinning or thickening, we computed the differences between seven different DEMs: a 1960s DEM from Berthier et al. (2010), a 1978 DEM from Dehecq et al. (2020), and the 2000, 2005, 2010, 2015, and 2020 DEMs from Hugonnet et al. (2021). Both the 1960s and 1978 DEMs are composites from different years (see Tab. B1 in App. B). The DEM by Berthier et al. (2010) was generated from historical USGS maps produced between 1948 and 1972, although for the glaciers of interest the information stems primarily from a single year (small portions in the accumulation area might stem from another year).

The DEM has a spatial resolution of 40 m and we will refer to it as the 1960s DEM. The DEM from Dehecq et al. (2020) is generated from declassified analog satellite stereo-images from the American reconnaissance program Hexagon (KeyHole-9) and covers the late 1970s. Here, the dates range from 1977 to 1980, but we will refer to this as the 1978 DEM. This DEM has a spatial resolution of 48 m. The DEMs from Hugonnet et al. (2021) cover the 21$^{st}$-century and have a spatial resolution of 30 m. These DEMs were created by temporal interpolation of several repeat DEMs either generated from stereo-images from the Advanced Spaceborne Thermal Emission and Reflection Radiometer (ASTER) or retrieved from the ArcticDEM strip archive based on WorldView stereo-images (Porter et al., 2018). At each pixel, on average 40 independent ASTER or ArcticDEM elevations acquired in the period 2000–2019 are used to predict elevation at five-year intervals between 2000 and 2020.

Prior to extracting the elevation changes from the DEMs, we co-registered them to the 2000 DEM to correct for shift and tilt misalignments. Horizontal and vertical shifts were removed following the iterative slope–aspect method of Nuth and Kääb (2011), then tilts were removed by fitting a 2-dimensional plane to elevation change residuals. Finally, we repeated the horizontal and vertical shift coregistration to remove sub-pixel shifts introduced by the tilt correction. This coregistration pipeline was performed using the xDEM package (xDEM contributors, 2024). The terrain used for coregistration was restricted to areas within 20 km of the glaciers of interest but excluding areas within 2 km of the RGI Consortium (2017) outlines. There was a shift of 2 m on average, consistent with the offset reported by Berthier et al. (2010). The vertical reference of the 1960s DEM had to be transformed from the EGM96 geoid to the WGS84 ellipsoid. The vertical shifts resulting from the transformation were between −10 and 20 m, depending on the location in Alaska.

We computed surface elevation changes over the following time periods: 1960–1978, 1978–2000, 2000–2005, 2005–2010, 2010–2015, 2015–2020. Unfortunately, elevation data is not available for Ellsworth and Columbia in 1978 so compared 1960 to 2000. We then examined the spatial distribution of the elevation changes throughout time, particularly over the glacierized area next to the landslide, and searched for periods of rapid thinning. The interpretation was aided by defining both a cross section and an instability-adjacent glacier (IAG) polygon (see Fig. 2 and App. C for descriptions). Using the cross section, we extracted point values from the seven DEMs to create elevation profiles. Using the IAG polygon, we computed the median elevation change over the glacierized area near the instability in different time periods.

### 3.3 Glacier terminus position

We used time series of glacier terminus positions mapped from Landsat images (1985–2022) to quantify glacier retreat and occasional glacier advance at yearly time steps. For Barry, Yale and Columbia, these data were already available until 2012 from McNabb et al. (2015), which provides manually-digitized terminus positions using all available Landsat images. For the other sites (Ellsworth, Portage, Tyndall, Alsek, and Grand Plateau), we manually mapped the terminus positions on an annual basis using cloud-free summer images to minimize snow cover. The vast majority of the images used were taken between July and October, though occasionally a winter image was used if no summer image was available. A list of images used for delineation is found in the Supplementary Materials.

We calculated retreat rates along the glacier centerlines, which were taken from RGI v6 (RGI Consortium, 2017). The centerlines had to be manually extended down-valley to account for larger glacier lengths during the start of our study period

(RGI outlines represent the glacier states around the year 2000). In some cases we manually refined the automatically generated RGI centerlines to more accurately represent the glacier and valley shapes. Finally, we calculated retreat by intersecting the centerlines with the glacier terminus positions, and cumulating the retreat relative to the largest glacier extent.

## 3.4 Supporting glacier and environmental data

### 3.4.1 Ice thickness

We used the two global ice thickness datasets by Farinotti et al. (2019) and Millan et al. (2022) to determine the valley topography below the glacier, put ice thinning rates into context, and estimate the ice remaining in front of the landslides. Rather than rely on a single product, we used both to obtain a range of possible values. Both Farinotti et al. (2019) and Millan et al. (2022) derive ice thickness from surface characteristics such as elevation, slope, and ice velocity. Farinotti et al. (2019) ice thickness dates refer to the glacier outlines from RGI Consortium (2017) version 6, while Millan et al. (2022) corresponds to the years 2017–2018.

The elevation of the glacier bed was determined by subtracting the given ice thicknesses from a 2015 DEM (Hugonnet et al., 2021). The usage of a DEM that does not correspond to the year of the ice thickness introduces some error, but we expect this to be minor as compared to the ice thickness uncertainty. For both datasets, we used the cross section (App. C) to extract the bedrock elevation and the IAG polygon (App. C) to compute the median thickness.

### 3.4.2 Meteorology

To determine the importance of non-glacier related environmental changes, we analyzed time series of precipitation. However, determining an appropriate time scale for this is challenging, as landslides can be activated by both heavy, short-duration precipitation events as well as extended periods of particularly wet conditions. We used time series of monthly temperature and precipitation from ERA5 reanalysis data (Hersbach et al., 2018) to determine how meteorological changes may correlate with landslide activity. For each site, we selected the grid cell (0.25° x 0.25°) encompassing the instability and retrieved the monthly values from 1979 until 2022. We averaged temperatures and summed precipitation totals to get annual values.

We inspected the data for changes in meteorological conditions which may explain increased slope activity, such as particularly rainy months or years. To do so, we computed the annual precipitation anomaly against a reference time period from 1980 to 2009 and evaluated this against the mean annual air temperature (MAAT) to identify potential "wet and warm years" that could have provided particularly high water inputs. We looked for years in which anomalously high precipitation correlated with increased landslide movement. Because the yearly time scales only provide a crude assessment, we also determined the average and standard deviation of monthly precipitation and investigated if periods of increased slope activity corresponded to times with anomalously high precipitation at the monthly level (exceeding two standard deviations above the mean).

### 3.4.3 Seismology

We used the USGS Earthquake Catalog to extract all seismic events in the study area between 1980 and 2023 (U.S. Geological Survey, 2023). We selected earthquakes which were within $100\,\mathrm{km}$ of a study site and that had a magnitude $\geq 4$. Magnitude 4 is considered to be the threshold at which earthquakes can cause landslides (Keefer, 1984). However, we acknowledge that pre-1980s events, in addition to smaller earthquakes, may have caused rock damage that could contribute to a slope destabilization.

We seek to quantify the effect of seismic activity on slope stability, since earthquakes may induce rock mass fatigue and promote failure (Gischig et al., 2016), though shaking can influence rock strength in a variety of ways, ranging from decreased to increased strength (Brain et al., 2021). We first calculate the energy $E$ released by each earthquake. For an earthquake with magnitude $M$, this is given by $E = 10^{(5.24+1.44*M)}$ (Earthquake Hazards Program, 2024). We then estimate an intensity $I$ of the seismic events at the location of the investigated instabilities, defined as the quotient of the earthquake energy $E$ and the square of the distance $d$ between the instability and the earthquake epicenter:

$$I(t) = \sum_t \frac{E(t)}{d^2}, \tag{1}$$

where $t$ is the time over one year. The summation thus represents the cumulative energy experienced at a site during a particular year. We inspected the resulting time series for periods where the seismic intensity increased sharply, and assessed whether such periods coincided with increased slope activity.

## 4 Results

To understand how glacier retreat controls landslide activity, we analyzed landslide displacements and glacier behavior from the early 1980s to present-day. Half of the studied landslides—Barry, Yale, Tyndall, and Grand Plateau—are currently situated adjacent to lakes or fjords which were occupied by glaciers in the 1980s. The other half of the landslides—Ellsworth, Portage, Columbia, and Alsek—still have contact with the glacier along the whole landslide toe. In the following, we first describe the most important stages of the landslide evolution at each site, and then describe the glacier activity during these stages. We summarize the temporal connections between the landslide and glacier changes, and present the findings of both the meteorological and seismic investigations.

### 4.1 Landslide evolution

Seven out of eight sites displayed clear down-slope movement between 1984 and 2022 (Fig. 3). At six of these sites, the combined use of ITS-LIVE data and manual feature tracking revealed distinct periods of acceleration or large surface changes (Fig. 4 and Fig. E1 in App. E). The landslides moved at average speeds of $\sim$2.8 to $\sim$8.7$\,\mathrm{m\,yr^{-1}}$ in the period 2000-2022 and up to a maximum velocity of $79\,\mathrm{m\,yr^{-1}}$ (Ellsworth). During periods of acceleration, the landslide velocities from ITS-LIVE increased up to a factor of 9 compared to the average velocity between 2000 and 2022. Manual feature tracking, on the other hand, gave maximum velocities up to 7 times higher than the long-term average (1984-2022). In the following, we will discuss the behavior of individual landslides in more detail.

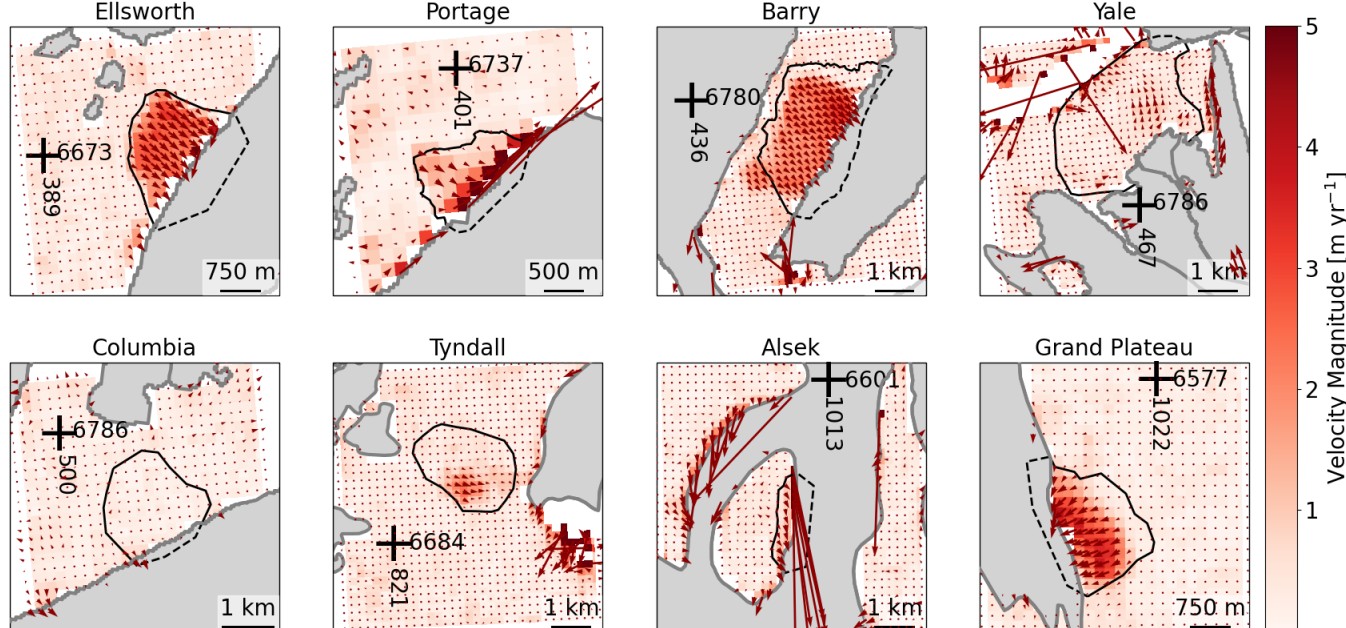

**Figure 3.** Average (1984-2022) displacement vectors and velocity magnitude derived from ITS-LIVE data. Glaciers are shown in gray (outlines from RGI Consortium, 2017) and black outlines mark the instability polygon (cf. Fig. 2). Coordinate crosses are given in UTM 6.

At Tyndall and Yale, landslide activity began as early as the 1980s and 1990s, respectively. At Tyndall, a 60 m-wide crack opened at the top of the landslide between 1987 and 1990 (Fig. 4f). Further disintegration of the slope followed, including
slope-scale motion starting in the 2000s, culminating in a catastrophic failure in 2015. Around 60 Mio. $m^3$ fell into the fjord and another 100 Mio. $m^3$ remained on the slope and have continued moving since 2015. At Yale, manual feature tracking showed that the landslide experienced a sudden pulse of movement between 1989 and 1995 (Fig. 4d), with surface features being displaced by around 75 m. The location of these surface displacements coincides with the region of highest velocity in the ITS-LIVE data (Fig. 3). The manual feature tracking did not reveal any visible motion after this period, though the ITS-LIVE
data indicates a small velocity increase around 2010, which may suggest ongoing creep.

At four other sites (Barry, Grand Plateau, Alsek, and Ellsworth), activity started between 2000 and 2010. The Barry landslide accelerated between 2004 and 2010 (Fig. 4c). Manual feature tracking indicates the movement started between 2009 and 2010, while ITS-LIVE data shows the acceleration beginning after 2008 and the velocity peaking around 30 m yr$^{-1}$ between 2010 and 2012. Between 2013 and 2016, the landslide slowed to around 10 m yr$^{-1}$ before the velocity stabilized at approximately
2 m yr$^{-1}$ in 2017. This acceleration displaced large parts of the landslide by around 200 m between 2004 and 2021.

During the same time (beginning of the 2000s), a $\sim$45 m-wide crack opened at the top of the Grand Plateau landslide (Fig. 4h). Manual feature tracking narrows down the movement onset to between 2007 and 2009. ITS-LIVE data show that the slope moved at a constant velocity of around 2.5 m yr$^{-1}$ between 2010 and 2022, though manual feature tracking indicates

much larger displacements. We suspect that this discrepancy is due to the slope appearance changing drastically and significant vertical motion, both of which pose challenges for the feature tracking algorithm.

At Alsek Glacier, the ITS-LIVE landslide velocity peaked at around $10\,\mathrm{m\,yr^{-1}}$ in 2008 and a $60\,\mathrm{m}$ wide crack opened between 2006 and 2008 (Fig. 4g). After this event, the landslide velocity decreased to around $\sim5\,\mathrm{m\,yr^{-1}}$ between 2010 and 2021. A similar phenomenon is observed at Ellsworth Glacier: The landslide accelerated dramatically in 2009, reaching velocities of around $80\,\mathrm{m\,yr^{-1}}$ and a cumulative displacement of $100\,\mathrm{m}$ between 2005 and 2010 (Fig. 4a). Later, between 2014 and 2021, the landslide continued to move, giving displacements around $30\,\mathrm{m}$.

Finally, at two sites, we detected limited or no movement over the study period. At Portage, the manual feature tracking did not reveal any motion (Fig. 4b) although ITS-LIVE data indicates clear down-slope motion (Fig. 3). This suggests that there is slow but possibly continuous motion without any sudden, large changes. Indeed, looking at higher-resolution aerial imagery, substantial disintegration of the slope between 1996 and 2006 is visible (see Fig. E2 in App. E). At Columbia, no motion could be detected by ITS-LIVE (Fig. 3) or by manual feature tracking (Fig. 4e), despite the fact that the landscape shows signs of deformation.

## 4.2 Glacier evolution

All glaciers retreated between 1984 and 2022, with total retreat varying from 0.97 to $23.4\,\mathrm{km}$ and average retreat rates ranging from 30 to $630\,\mathrm{m\,yr^{-1}}$. Yale and Tyndall experienced short periods of advance, but all the 2023 terminus positions are up-valley of the 1980s extents. In accordance with the observed retreat, all sites thinned between 1960 and 2020, with average thinning rates ranging from 1.8 to $9.6\,\mathrm{m\,yr^{-1}}$ (Tab. F1 in App. F, as well as Fig. 5). At four out of eight sites, periods with rapid glacier changes coincide with periods of landslide activity and slope acceleration (Fig. 4), which we focus on in the following.

Between 1984 and 1988, Tyndall Glacier retreated up-fjord by $5\,\mathrm{km}$, and the feature tracking shows that the head scarp crack opened as the glacier terminus passed the landslide. After this dramatic retreat, the glacier terminus stabilized, fluctuating slightly around its new position. In 2021, it was around $800\,\mathrm{m}$ longer than its 1990 extent. At the terminus, the glacier thinned by $6.4\,\mathrm{m\,yr^{-1}}$ between 1960 and 2000, and thickened by $0.6\,\mathrm{m\,yr^{-1}}$ between 2000 and 2020 (Tab. F1; Fig. 4f). The glacier elevation near the landslide is currently around $200\,\mathrm{m\,a.s.l.}$, around $200\,\mathrm{m}$ less than in the pre-2000s period.

Aside from Tyndall, Yale is the only other glacier that experienced short phases of advance. Here, the terminus advanced by about $130\,\mathrm{m}$ between 2004 and 2014 (Fig. 4d). Despite these short phases of stability, Yale Glacier retreated by around $1\,\mathrm{km}$ since the 1980s, a trend that was accompanied by rapid thinning. On average, Yale glacier thinned by $3.3\,\mathrm{m\,yr^{-1}}$, the thinning rate being more than twice as high before the year 2000 than afterwards ($5.5\,\mathrm{m\,yr^{-1}}$ versus $2.2\,\mathrm{m\,yr^{-1}}$; Tab. F1). The increased thinning rate before 2000 also coincided with one of the most rapid periods of retreat: between 1989 and 1995, the glacier terminus retreated past the landslide area at around $70\,\mathrm{m\,yr^{-1}}$, which is more than twice the long-term retreat rate of $30\,\mathrm{m\,yr^{-1}}$. Just as at Tyndall, the strong retreat coincided with the main landslide displacement.

Barry Glacier retreated most rapidly up-fjord between 2003 and 2016, and slope motion began between 2009 and 2010 when the terminus was adjacent to the down-valley margin of the landslide. During this period, the retreat-rate peaked around $230\,\mathrm{m\,yr^{-1}}$, nearly three times higher than the average retreat rate over the whole 1985–2022 period. In 2016, the terminus

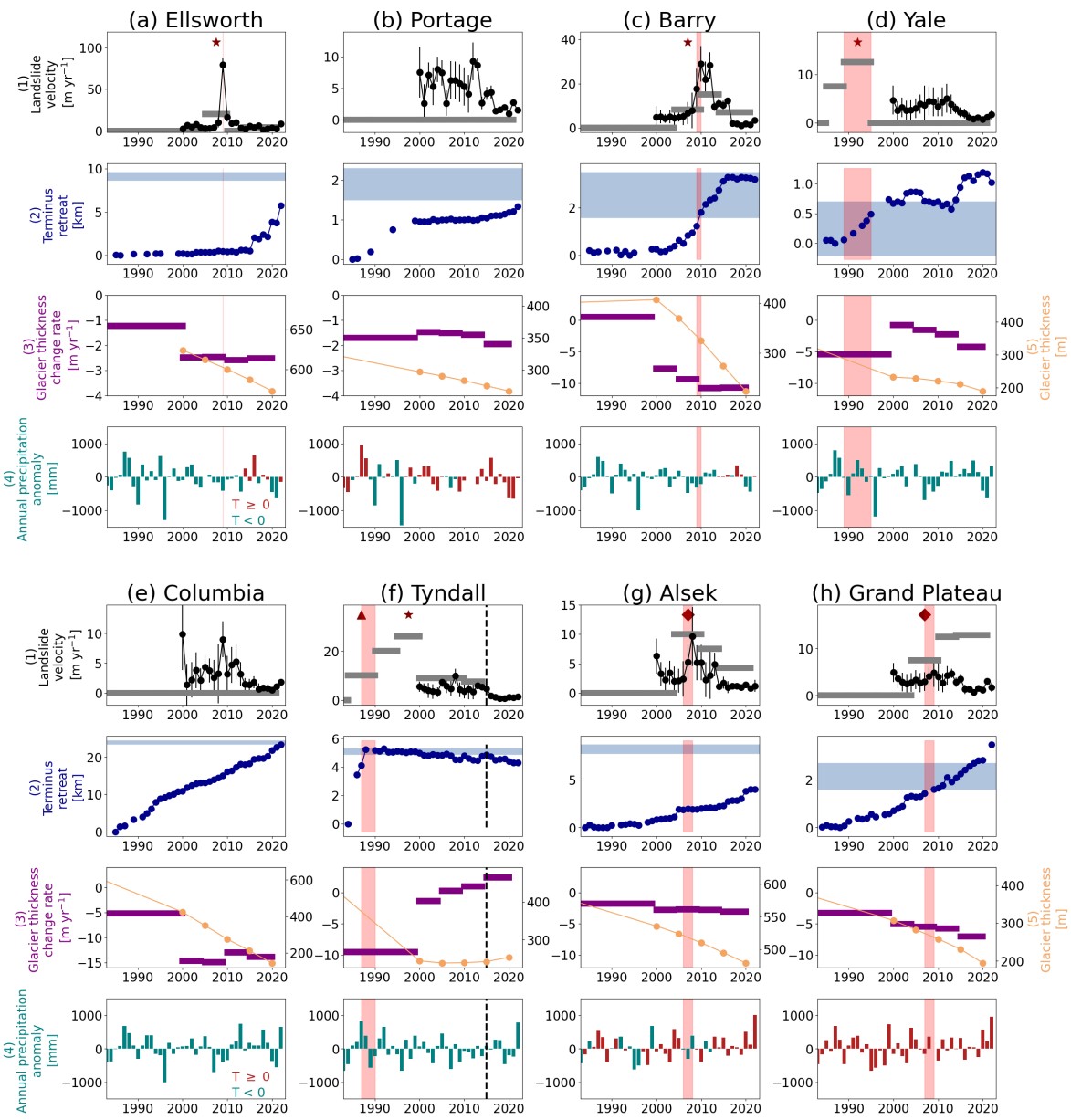

**Figure 4.** Landslide and glacier evolution at the study sites. Row 1: Landslide velocities from ITS-LIVE (black circles, with uncertainty estimates) and manual feature tracking (gray bars). Stars indicate onset of slope-wide deformation, triangles stand for crack opening, and diamonds mean both deformation and crack opening. Row 2: Terminus retreat (dark blue) and location of the landslide along the glacier centerline (light blue shading). Row 3: Glacier thickness change rates (purple) and absolute ice thickness (yellow; right hand axis) below the landslide. Row 4: annual precipitation anomaly from ERA5 (Hersbach et al., 2018) (relative to 1980–2009), where teal and red bars indicate a positive or negative MAAT. In all panels, light red shading indicates the onset of landslide movement. At Tyndall, the black dashed line indicates the failure. Note the differing scales on the y-axes.

position stabilized at its current location near the up-valley end of the landslide, having retreated past $\sim75\%$ of the landslide toe. Deformation of the terminus in response to the landslide movement is clearly visible from the ITS-LIVE velocity vectors (Fig. F1 in App. F). Aside from some mild thickening prior to 2000 ($0.9\,\mathrm{m\,yr^{-1}}$), Barry Glacier has experienced the second highest thinning rate of all study sites, thinning by $9.7\,\mathrm{m\,yr^{-1}}$ per year between 2000 and 2020 (Figs. 4c and 5). Since 2000, the glacier has lost around $350\,\mathrm{m}$ of ice near the landslide.

Similar to Barry, Grand Plateau Glacier retreated around $3\,\mathrm{km}$ between 1985 and 2022, and the current terminus position lies up-valley with respect to the landslide. Like Barry, slight deformation of the terminus in response to the landslide is visible (Fig. F1 in App. F). Thinning rates have been generally increasing since the 1980s (Fig. 4h). Near the landslide, the glacier thinning rate was $2.6\,\mathrm{m\,yr^{-1}}$ on average before 2000, and increased by a factor of two between 2000 and 2020. Since the 1980s, the retreat rate of Grand Plateau gradually accelerated as the glacier moved up-valley through the lake, but there was not one distinct period of acceleration. The glacier retreated around $90\,\mathrm{m\,yr^{-1}}$ on average between 2007 and 2009, coinciding with both the glacier passing the down-valley margin of the landslide and the opening of the crack (Sect. 4.1).

At Ellsworth and Alsek glaciers, both retreat and thinning rates have increased over time (Fig. 4 and 5). Next to the landslide, Ellsworth Glacier's thinning rate was $1.2\,\mathrm{m\,yr^{-1}}$ prior to 2000 and increased by a factor of two between 2000 and 2020, corresponding to the time when the landslide accelerated. The terminus position was fairly stable in a proglacial lake until 2014, when the tongue of the glacier began disintegrating and retreat rates increased. Currently, the instability is still around $3\,\mathrm{km}$ from the terminus and as much as $250\,\mathrm{m}$ of ice buttresses the landslide. At Alsek, the glacier adjacent to the landslide thinned by $1.6\,\mathrm{m\,yr^{-1}}$ between 1960 and 2000, and by $2.8\,\mathrm{m\,yr^{-1}}$ between 2000 and 2020, a factor of 1.8 higher. This brackets the period with the crack opening. Similarly, Alsek Glacier retreated around $45\,\mathrm{m\,yr^{-1}}$ prior to 2000 and this rate increased by a factor of three since then. However, the terminus is still about $4\,\mathrm{km}$ down-valley from the instability.

Both Portage and Columbia thinned during the study period, and the terminus is currently several hundred meters from the instability centerpoint (Fig. 4). Along with Tyndall and Yale, Portage Glacier is one of three sites where the thinning rate was higher before 2000 than it was afterwards: Portage thinned at $2.2\,\mathrm{m\,yr^{-1}}$ before 2000, and at around $1.6\,\mathrm{m\,yr^{-1}}$ between 2000 and 2020. The glacier retreated rapidly ($65\,\mathrm{m\,yr^{-1}}$) through the proglacial lake between 1985 and 2000, remained stable for around 15 years, and retreated at around $40\,\mathrm{m\,yr^{-1}}$ since 2015. Columbia Glacier, on the other hand, thinned by $5.2\,\mathrm{m\,yr^{-1}}$ between 1960 and 2000, increasing by a factor of 2.7 to $14.1\,\mathrm{m\,yr^{-1}}$ between 2000 and 2020. Here, steepening of the slope, likely due to glacier erosion, is clearly visible below the 1960s glacier surface (Fig. 5). Between 1985 and 2022, the glacier retreated up-fjord by over $23\,\mathrm{km}$, or a rate of around $630\,\mathrm{m\,yr^{-1}}$.

## 4.3 Other environmental controls

The results presented in the sections above suggest that glacier changes, especially rapid ones, may exert a strong control on landslide acceleration. To make this inference, however, alternative factors that might have initiated the larger slope displacements should be ruled out. Below we address the possible influences of precipitation and seismicity.

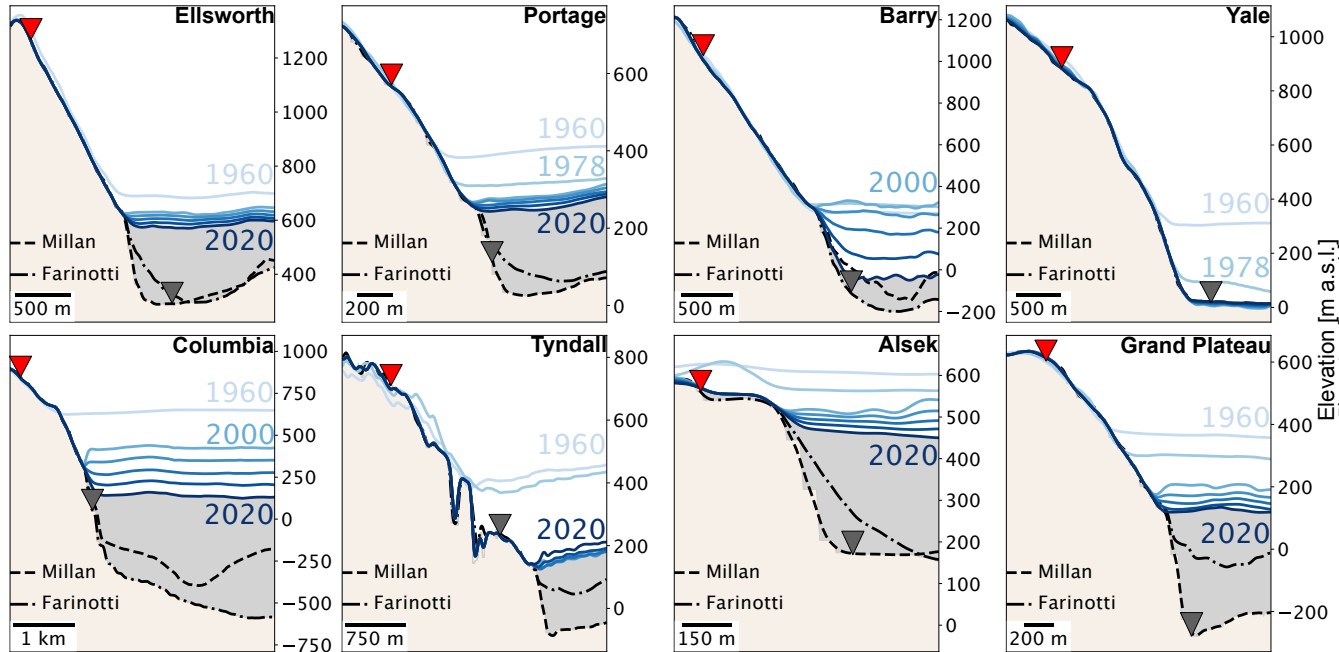

**Figure 5.** Cross sections through the landslide and glacierized area at each site. Bedrock (tan) and glacier areas (gray) are shown. Glacier surface elevations are plotted in progressively darkening shades of blue. Subglacial topography is given by two ice thickness datasets (Millan et al. (2022) and Farinotti et al. (2019)). Red and gray triangles mark the upper and lower bounds of the landslide, respectively. From the two bedrock estimates, the gray triangle was arbitrarily placed on the Millan surface. Note the horizontal and vertical scales vary between sites.

### 4.3.1 Meteorology

We examined annual precipitation anomalies at all sites (Fig. 4). At most sites, precipitation totals during the year(s) of the landslide acceleration were close to the long-term average. At Ellsworth, Barry, Yale, Alsek, and Grand Plateau, small positive or negative precipitation anomalies were observed during the year(s) of the movement onset. However, in none of these cases were the anomalies exceptionally high when compared to the rest of the time series. At Tyndall, the year 1987 was characterized by a precipitation total over $800\,\text{mm}$ above the long-term average—the largest anomaly in the time series. During this period of landslide activation, there were two years with above-average precipitation, followed by two years with below-average precipitation. Due to limited good-quality image availability in the 1980s, there is large uncertainty in the exact onset of the landslide movement and thus the impact of the consecutive wet years on landslide initiation cannot be ruled out.

In addition to the annual time scale, we analyzed monthly precipitation totals at the six sites where landslide accelerations were observed (Figs. G1-G3 in App. G). At Ellsworth, where the landslide accelerated between June and August 2009, yearly precipitation was below average (Fig. 4a). Amidst this dry period, however, the precipitation during July 2009 was 80% above the average July precipitation ($293\,\text{mm}$). A connection between precipitation and landslide acceleration can thus not be ruled out in this case. At Alsek and Grand Plateau, some monthly totals during the landslide activation period were above average,

but never exceeded two standard deviations above the mean. In the cases of Barry, Yale, and Tyndall, certain months did have anomalously high precipitation during the period of the landslide activation. However, because we cannot determine the onset of the landslide movement more precisely, it is unclear if the accelerations might be related to these large precipitation amounts or if they happened during a different time of the year.

### 4.3.2   Seismicity

The seismic data were scrutinized to determine whether slope accelerations followed intense or prolonged seismic activity (Fig. H1 in App. H). We did not observe a temporal correlation between high seismic intensity and landslide velocity during the years of landslide activation, nor did we observe increased landslide activity in the years following a particular seismic event. While we acknowledge that seismic shaking can cause rock damage which impacts landslide stability (see Sect. 5.2), the evidence here shows no direct link between specific seismic events and landslide acceleration.

## 5   Discussion

The impact of glacier thinning on landslide destabilization has been investigated at several locations across the world, and there is general agreement that glacier thinning contributes to landslide destabilization (e.g., McColl and Davies, 2013; Glueer et al., 2019; Cody et al., 2020; Lacroix et al., 2022). So far however, most studies investigating glacial debuttressing have taken place at sites where the glaciers were land-terminating. Observations of landslides adjacent to retreating glaciers which give way to water are much less common (an exception is given by the studies of Kim et al. (2022) and Dai et al. (2020)), as is documentation of accelerations at such sites.

In this study, we investigated the response of paraglacial landslides to glacier changes over multiple decades. Our results show that at Yale, Tyndall, Grand Plateau and Barry—all of which are adjacent to lakes or fjords—the slope-wide deformation began when the glacier retreated past the instability. Ellsworth experienced an increase in glacier thinning, but the landslide acceleration also coincided with an anomalously rainy month. At Alsek, we found a sudden activation which was preceded by an increase in glacier thinning. Finally, despite the continuous mass loss of both glaciers, Portage showed very little movement without a clear period of acceleration, and at Columbia, no movement was detected. In the following, we discuss possible differences between the land- and water-terminating situations at varying stages of retreat and evaluate the possible slope evolution following destabilization. We also compare our results to previously published work and propose future research perspectives.

### 5.1   Landslide evolution in land- vs. water- terminating situations

As glaciers retreat, the hydrology and mechanics of the paraglacial slopes must adjust accordingly (Fig. 6). In temperate glaciers (ice at or near 0°C), water is readily available and it may flow on the surface, within, and below the glacier (Jansson et al., 2003). Englacial water storage can impact the pore pressure in neighboring slopes on seasonal timescales (Grämiger et al., 2020; Hugentobler et al., 2020), meaning that adjacent to the glacier, liquid water is present in subsurface pores and cracks and

### a) glaciated conditions

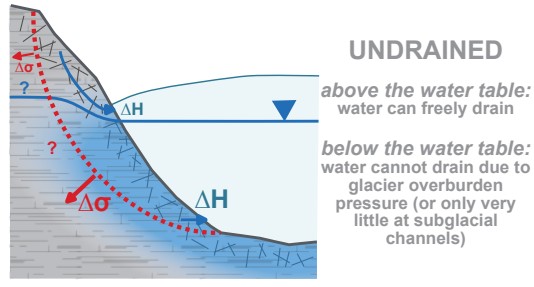

### b) during de-glaciation

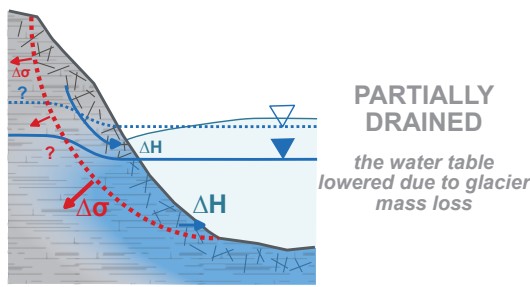

### c) upon glacier disappearance

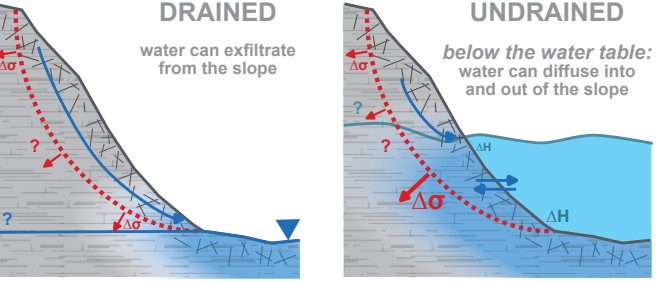

**Figure 6.** Conceptual figure outlining the changes to slope hydrology during different stages of glacier retreat (panels A and B), as well as in the presence of a proglacial lake or fjord after glacier disappearance (panel C). Rock mass is gray, glacier ice is light blue, and lake/fjord water is blue. The red line is the assumed failure plane of the landslide, and red arrows represent the slope displacement. The solid blue line and filled triangle indicate the current water table. The dashed blue line and unfilled triangle indicate the former water table. Blue arrows indicate where water can drain. "$\Delta H$" is the overburden pressure from ice or water, and the size is representative of the magnitude. "$\Delta\sigma$" is the pressure on the failure surface and the size represents the magnitude. The terminology of "drained" and "undrained" conditions follows Lacroix et al. (2020).

cannot drain due to the ice overburden pressure (Fig. 6a). The presence of water in the subsurface can be problematic for slope stability by decreasing the frictional forces and increasing pore water pressure within the slope, both of which contribute to instability (Blasio, 2011; Lacroix et al., 2020). Existing work has shown that during ice sheet retreat, groundwater can exfiltrate from the subsurface due to decreasing ice overburden pressure (Ravier and Buoncristiani, 2018). Thus, as the glacier surface

lowers, the water table in the slope moves to a correspondingly lower elevation (Fig. 6b) and the upper portion of the slope
becomes "drained," a term defined by Lacroix et al. (2020). In this case, excess pore water pressure cannot develop and this
portion of the slope becomes stabilized. The part of the slope still under the water table, on the contrary, would remain saturated
and "undrained."

Upon complete disappearance of the glacier, the balance between the various other external forces (e.g. hydrological, mechanical, and geological) determines if the slope is stabilized or begins to move. The slope may be completely drained—in
the case of a land-terminating landslide—or undrained in the case of a water-terminating one (Fig. 6c). In the former case, a
new, stable equilibrium may be found as the slopes (re-)adjust to the changed boundary conditions. In the latter case, the slope
remains saturated below the lake or fjord level and, similar to the glaciated case, hydraulic gradients can cause pressure changes
along the failure surface, which impact slope stability (McColl, 2015). Such fluxes and pressure changes may cause further
damage to rock which has already been weakened by glacier retreat and thinning (Grämiger et al., 2017, 2018). Additionally,
the lower portion of the slope must support the mass above it, which it may not be able to do effectively when saturated. Thus
in the case that an ice buttress is replaced by water, a precarious situation exists where a saturated and weakened subsurface is
loaded by the mass above it, though it remains unknown how this may increase the chances for future slope failure.

Such groundwater changes during glacier retreat have been found to be a critical element causing slope destabilization (Bovis
and Stewart, 1998; Oestreicher et al., 2021). Grämiger et al. (2020) suggested that glacier changes at the toe of a landslide may
resemble lake level fluctuations in the subsurface of the adjacent slopes. Changing lake levels, and particularly reservoir filling
and drawdown cycles, have been shown to drive instability, as does the rate at which these cycles occur (Paronuzzi et al., 2013).
The hydrological fluctuations driven by ice loss and the subsequent infilling of a proglacial water body, may mimic falling and
rising lake levels and promote instability. Given that glacier mass loss is particularly rapid in southern Alaska (Hugonnet et al.,
2021), the pace of these glacially driven subsurface hydrological changes is likely also relevant.

In addition to groundwater changes, there are mechanical changes when an ice buttress is replaced by water. Although ice
can deform at very low strain rates (McColl et al., 2010) and has little influence on the slope stability (Storni et al., 2020), the
buttressing provided by ice is more pronounced than the one provided by water after glacier retreat. Additionally, the rapid
retreat rates of lake-terminating or tidewater glaciers mean that this change occurs relatively suddenly for landslides adjacent
to deep water. As calving events occur and give way to the vertical cliffs that are characteristic of calving fronts (Cuffey and
Paterson, 2010), slopes would experience drastic boundary changes in a short time frame. At Petermann Glacier, Greenland,
for example, a single calving event in August 2010 led to the loss of around $250\,\mathrm{km}^2$ of ice (Falkner et al., 2011). At Ilulissat
(Jakobshavn) Glacier, Greenland, calving events in 2007 and 2008 extended up to a kilometer horizontally and several hundred
meters up-glacier (Amundson et al., 2008). The abrupt loss of large ice pieces may lead to localized structural changes within
the landslide area (Grämiger et al., 2017; Hugentobler et al., 2022), which might cause additional, slope-scale redistribution of
mechanical stress as the landslide mass starts to deform.

## 5.2 Landslide interactions with the glacier and environment

Examining landslide and glacier evolution over several decades allowed us to determine the timing of what we refer to as landslide *activation*. It is possible that the underlying structures of these landslides have existed for many decades, centuries, or even millennia, and previous work has shown that landslides can reactivate or even fail millennia after deglaciation (Hermanns et al., 2017). We thus do not speak about landslide initiation, which would imply the motion onset, nor do we rule out that earlier phases of activity may have existed. We use the term *activation* to indicate the onset of detectable movement during our study period, while acknowledging that it may be a "reactivation" which we are observing.

When trying to establish a mechanism relating glacier changes to the observed instabilities, our sites can be divided into two categories: those where the glacier terminus has retreated beyond the landslide and has been replaced by water, and those where the terminus is still downstream of the landslide. We will refer to landslides in the former and latter cases as being impacted by "retreat-related" and "thinning-related" debuttressing, respectively. We acknowledge that glacier retreat and thinning are closely linked, meaning that the processes cannot be fully separated, but we use these terms to distinguish between slopes which have undergone a complete loss of glacier support from slopes in which some glacier buttressing is still present.

At Yale, Tyndall, Grand Plateau, and Barry, we observed a sudden activation of the landslides in response to rapid glacier retreat. In all four cases, the glaciers retreated past the landslides through lakes or fjords, and at rates up to seven times their long term average. For glaciers terminating in water, a combination of warm water intrusion (Weertman, 1974; Luckman et al., 2015), dynamic thinning (O'Neel et al., 2005; Benn et al., 2007), and surface melt (Warren et al., 2001; Benn et al., 2007) may cause the terminus to thin to the point that it floats in a proglacial body of water (Warren et al., 2001). With part of the glacier floating, there is less basal friction and the glacier velocity can increase, thus supplying more ice to the calving front, and further thinning it (Benn et al., 2007). Since the rate of calving tends to increase with water depth, retreat is fostered in deeper water (Hanson and Hooke, 2000; Benn et al., 2007). These feedback processes between ice dynamics and submarine topography cause the rapid retreat which is typical for marine- or lake-terminating glaciers.

In addition to the rapid retreat rate, there are other predisposing factors that may make these sites prone to acceleration. First, three out of four sites are in sedimentary lithologies (Yale, Tyndall, and Barry), which are particularly susceptible to water intrusion due to high porosity (Selley, 2005). Second, due to the maritime climate of southern Alaska and the low elevation (Sec. 2), the glaciers at our study sites are presumably temperate, meaning that there is high water availability in the subsurface prior to deglaciation. This combination of weakened and porous rock with high water availability may make these landslides particularly likely to accelerate as the boundary conditions change during especially rapid glacier retreat.

The above considerations suggest that landslides adjacent to rapidly retreating lake- or marine-terminating glaciers may be particularly susceptible to destabilization. However, landslide activity was also detected for sites that "only" experienced thinning. At Ellsworth and Alsek, for example, landslide activation occurred 6 to 8 years after glacier thinning increased by a factor of ~2. Lacroix et al. (2022) made a similar observation in Iceland, where a landslide acceleration at Tungnakvíslarjökull followed 6 years after a marked increase in glacier thinning rate. Kos et al. (2016) proposed that landslide activation may begin after the glacier thins to a critical thickness. For the Moosfluh landslide in Switzerland, for example, Storni et al. (2020)

found that slope displacements were larger where ice thickness was below $50\,\text{m}$ and smaller where the ice was ca. $100\,\text{m}$ thick, while Glueer et al. (2019) found that the whole landslide accelerated after the ice thinned below $100\,\text{m}$. Our results show that the landslide accelerations at Ellsworth and Alsek occurred when the local ice thickness was on the order of $300\,\text{m}$ (Ellsworth) to $350\,\text{m}$ (Alsek). However, these numbers cannot be directly compared to the Moosfluh case, which is an alpine glacier in very different climatic conditions, but may imply region-specific thinning thresholds. Taken together, thinning-related debuttressing does not seem sufficient to explain all of the observed landslide behavior, and indeed, the Portage landslide deformed continuously as the glacier thinned, and no evident acceleration was detected at Columbia despite an increase in glacier thinning rate by almost a factor of three.

Precipitation might be a further cause of landslide activation, in addition to glacier changes. Here, we chose to investigate temporal resolutions that are consistent with the available information on glacier and landslide changes, acknowledging that by doing so we cannot capture the full variability of the meteorological processes. Existing literature shows that landslides can respond to a wide range of precipitation totals, both in wet and dry climates (Handwerger et al., 2022), and on timescales ranging from minutes to several months (Iverson, 2000; McColl, 2015). Similarly, there is evidence that interannual precipitation changes play a role, with a number of landslides being initiated during rainy years following a drought (Handwerger et al., 2019a). Lacroix et al. (2020) states that elevated landslide velocities are typically seen during i) periods with intense precipitation, ii) times with long-lasting precipitation, or iii) the melt season, but that landslide response to precipitation also depends on the local geology, the specific characteristics of the instability, as well as the local topography and hydrology.

In the case of Ellsworth, we found above-average July precipitation coinciding to landslide acceleration, although the precipitation total in July was lower than typical autumn amounts. This indicates that the total precipitation was not unusual for the site, while the timing was. This could be relevant for two reasons. First, the available temperature data indicate that the precipitation likely fell as rain in July, as opposed to snow in autumn (Fig. G1 in App. G). Second, the subsurface may have been additionally saturated from ice melt in summer. Together, these conditions could have led to above-average pore water pressure and thus, to the observed landslide acceleration. Being based on observations of one site only, however, we recognize that such a mechanism remains speculative.

Also the relation between seismic activity and landslide activation remains inconclusive. Our study period began in 1980, 16 years after a magnitude 9.2 earthquake rocked Alaska (Kanamori, 1977), and it is not excluded that such a large seismic event might have weakened the various slopes through mechanical damage without causing instantaneous failure. This is in line with findings by McColl (2015), who noted that i) landslides can withstand many dynamic events (such as due to seismic activity) without failing and ii) the event triggering catastrophic failure may not be the largest. In an analysis focusing on central Italy, for example, Song et al. (2022) found that landslide accelerations occurred in areas with light-to-moderate ground-shaking. While we do not find a direct link between seismic activity and slope accelerations in our data, we recognize that seismic events can contribute to the development of instabilities through a preconditioning of the related slopes.

There are additional factors which, while relevant for slope stability, did not play a central role in this analysis. For example, the site-specific structural geology, such as the composition of the landslide and the fracturing of the subsurface, is undoubtedly a critical factor for slope stability. The subsurface properties cannot be determined without intensive field studies, but

such studies are not feasible at the scale studied here. Similarly, we did not consider any specific information on the slopes' hydrology, as such information is simply not available. Plus, it is closely linked to subsurface fracturing, which cannot be determined from remote sensing. Slope hydrology is also influenced by seasonal melt events. Annual snowpack releases large amounts of water into the slope, which may impact stability. Detailed analyses on snow hydrology are outside the scope of this work, but an analysis of total solid precipitation over the time period 1980–2022 did not show a correlation with landslide activation (Fig. G3 in App. G).

Our results suggest that paraglacial landslides—especially those that are in contact with a lake or fjord after glacier retreat—may respond rapidly to glacier mass loss. However, the initial landslide response to does not determine the long-term evolution of the landslide, where both a re-stabilization or catastrophic failure might occur. At Yale, for example, no further large-scale movement followed after the landslide activation in the early 1990s. On the other hand, at Barry and Grand Plateau, measurable deformation continued after activation. Barry accelerated rapidly and then slowed, while Grand Plateau moved at a constant, accelerated pace. While slow movement may continue for a long period of time, landslides can experience periods of acceleration and some even progress to catastrophic failure (Lacroix et al., 2020), with Tyndall being an example of the latter. The reason why some landslides maintain a slow velocity and others speed up or fail is unknown, but slow motion is typically observed prior to failure (Hendron and Patton, 1987; Handwerger et al., 2019b; Federico et al., 2011). This transition from slow to fast movement may be related to decreasing porosity in the shear-zone (Agliardi et al., 2020; Iverson et al., 2000), decreasing viscosity of the landslide material (Mainsant et al., 2012; Carrière et al., 2018), or shear localization (Voight, 1988; Lacroix and Amitrano, 2013) and is a topic of ongoing research.

### 5.3 Comparison to previous works

Other work at five of the eight sites has independently confirmed landslide movement. Schaefer et al. (2024) found average line-of-sight speeds ranging from $0.41$ to $9.64\,\mathrm{mm\,yr^{-1}}$ for Ellsworth, Portage, Barry, Yale, and Columbia. These values are very different from the ones obtained in this work, which we attribute to the differing methods and different periods of investigation. While Schaefer et al. (2024) used InSAR data, which measures small displacements but may not be suitable for large accelerations (Manconi, 2021), we employed lower resolution satellite data. Additionally, Schaefer et al. (2024) investigated the period 2016-2022, and we characterized the long-term evolution over a 40-year period. Of the 43 sites investigated by Schaefer et al. (2024), 11 were determined to be potentially tsunamigenic, and four of those 11 sites are also studied here. Regardless of the drivers, landslide movement and accelerations near deep lakes or fjords have important hazard implications.

In-situ measurements at Portage and Columbia have also confirmed movement at those sites. Deformation up to $5\,\mathrm{m}$ at the Portage landslide is visible in high-resolution digital image correlation between 2022 and 2023 (Lemaire et al., 2023b). This is higher than the $2.1\,\mathrm{m\,yr^{-1}}$ detected by ITS-LIVE on average over 2021–2022 (Fig. 4f). At Columbia, movement on the order of a few centimeters per year has been detected using GPS (Jeffries, 2023) and InSAR (see Supplementary Materials). The discrepancy between these numbers and our results can be explained by the resolution of the satellite imagery (see Sect. 3.1).

At Barry, Dai et al. (2020) characterized the landslide movement since 2000 using feature tracking. A comparison to the ITS-LIVE results shows generally good agreement in terms of both timing and average velocity magnitudes, the largest differences

**Table 1.** Comparison of landslide velocity at Barry Glacier from Dai et al. (2020) and this work. The 'Ratio' is computed by dividing the value from this work by the value from Dai et al. (2020).

| Time Period | Dai et al. (2020) [$\mathrm{m\,yr^{-1}}$] | This Work [$\mathrm{m\,yr^{-1}}$] | Ratio [–] |
|---|---|---|---|
| 1999 - 2008 | $1.3 \pm 0.6$ | $6.4 \pm 4.8$ | 4.9 |
| 2010 - 2013 | $26.2 \pm 3.0$ | $22.2 \pm 6.0$ | 0.8 |
| 2014 - 2016 | $9.6 \pm 2.0$ | $11.3 \pm 1.7$ | 1.2 |
| 2017 - 2020 | $1.3 \pm 0.7$ | $1.7 \pm 0.2$ | 1.3 |

being found in the period 1999-2008, when the uncertainty is largest, and the smallest differences being found in the periods 2010-2013 and 2014-2016 (Tab. 1). In their analysis, Dai et al. (2020) used imagery from Landsat (as we did), but also from ASTER, WorldView-1, and Ikonos, which have partly much higher resolution. This may explain some of the discrepancies.

### 5.4 Future research perspectives

Our study suggests that the rapid retreat of lake- and marine-terminating glaciers can lead to the sudden activation of paraglacial landslides near bodies of water. Since the retreat rate in such cases is related to water depth (Sect. 5.2), this raises the question whether landslide activation is preferentially co-located with deeper-than-average sections of lakes or fjords. Accurate bathymetric data would help answer this question, as they provide a detailed picture of the submarine environment adjacent to the landslides. If such a relation between water depth and landslide activation exists, ice thickness datasets (Farinotti et al., 2019; Millan et al., 2022) could be used to estimate the up-valley extent of the lakes or fjords and thus potential future water depths. This would be informative from a hazards perspective, also because the velocity of a potential tsunami triggered by a landslide collapsing into water is known to be proportional to water depth (Okal, 1988).

In terms of mechanistic understanding of the processes at play during landslide activation, the four cases where rapid glacier retreat coincided with landslide activation are obviously not sufficient to establish conclusive causalities. Broader regional studies with a focus on paraglacial landslides adjacent to lake- or marine-terminating glaciers would be helpful in this respect. At a more local level, detailed observations at sites where the glacier terminus is projected to soon pass the landslide area (e.g. Portage and Columbia) could yield valuable insights into the changing boundary conditions. Together, such regional and local studies could also help in further testing our framework distinguishing retreat- versus thinning- dominated debuttressing.

As glaciers continue to retreat and expose new fjords and lakes, the proximity of instabilities to water will change. This will have consequences in terms of hazard disposition and possible mitigation measures. We argue that additional observations would be useful in order to monitor known landslides and detect newly forming instabilities in a timely manner. Together, this may help to minimize the risk that a rapidly evolving environment poses to the public.

# 6 Conclusions

This work provides a comparison of several glacier debuttressing-related instabilities in southern coastal Alaska. We studied eight large landslides which are currently in contact with a glacier or have been so in recent decades, and which show signs of recent activity. At all sites, we use feature tracking in combination with glaciological, meteorological, and seismological datasets to examine correlations between slope movement and environmental changes between the 1980s and present-day. To extract slope velocities, we primarily use the ITS-LIVE dataset. While this dataset has fairly coarse resolution ($120\,\mathrm{m}$) and was originally designed for quantifying glacier flow velocities, comparison with in-situ observations (Dai et al., 2020) attest the suitability of the dataset for our purposes (Sec. 5.3).

We find that six out of the eight sites underwent significant slope acceleration at some stage during the studied time period. At four sites, such an acceleration occurred as the glacier terminus retreated past the landslide area. In these cases, the landslides border deep water bodies and we suggest that they underwent rapid debuttressing as the glacier retreated up-valley. For another two sites, landslide acceleration either coincided with a particularly rainy month, or with a significant increase in glacier thinning. The remaining two sites instead showed either slow, constant movement without a specific period of acceleration, or no detectable movement at all. In terms of causality, we suggest that the landslide accelerations were related to a loss of mechanical support from the glacier and changes in the landslide hydrology.

The presence of large, unstable slopes poses a significant hazard, particularly when located in the vicinity of deep water. The rapid ongoing glacier retreat could expose more such slopes in the future, potentially increasing the risk for some of these to fail catastrophically. In this context, we see two potential avenues for further work: First, more detailed investigation of the sites would be enlightening, since only limited information can be gained from remote sensing methods. In-situ monitoring, for example, would provide more reliable data than available via the ITS-LIVE results, as well as a more complete picture of the local processes. This is particularly important at sites that will experience debuttressing in coming decades, such as Portage or Columbia. Second, we see potential in using the ITS-LIVE data for detecting landslide events at a larger, potentially worldwide, scale. Some aspects, such as the detection limits of the method and the leakage of the glacier signal to neighboring areas, would need to be addressed for that, but could lead to the early detection and monitoring of landslides at the regional scale. Such a regional overview would allow for correlating landslide activity with various factors over a broader area, specifically elevation, aspect, precipitation amount, and proximity to faults, to name a few. In the longer term, such a development could assist in dealing with the hazards that stem from a rapidly changing environment.

*Code and data availability.* Please contact the first author regarding the availability of code and data used in this work.

*Video supplement.* Videos showing the landslide activation at four sites are available as supplementary material.

 **Appendix A: Site Description**

## A1 Ellsworth

Ellsworth Glacier is a lake-terminating glacier located on the Kenai Peninsula, around $30\,km$ east of Seward, Alaska (Fig. 1B and Fig. 2). It is oriented to the southwest and flows from a glacier complex at over $1800\,m\,a.s.l.$ to nearly sea level ($10\,m\,a.s.l.$). There are a series of instabilities along the western edge of the glacier, where the glacier bends. We focus on the
instability which is largest and farthest up-glacier, spanning a distance between $2.5$ and $4\,km$ from the 2021 terminus. The instability is characterized by a prominent main scarp and an active talus source area (Higman et al., 2023). Movement at the site between 2016 and 2022 has been confirmed by Schaefer et al. (2024) using interferometric synthetic aperture radar (InSAR). The instability is located in sedimentary rocks of the Orca group (Eocene to Paleocene age) and the lithology of the group is sedimentary, consisting primarily of sandstone and siltstone (Wilson et al., 2015). The volume is estimated between
$66$ and $150\,Mio.\,m^3$ ($14$ to $113\,Mio.\,m^3$ according to Schaefer et al. (2024); Tab. D2 and App. D).

## A2 Portage

Portage Glacier is a lake-terminating glacier located at the northern part of the Kenai Peninsula, around $7.5\,km$ southwest of Whittier, Alaska (Fig. 1A and Fig. 2). The glacier flows northeast and extends from an ice field at $1430\,m\,a.s.l.$ to around $60\,m\,a.s.l.$. At the north end of the glacier, where the glacier calves into Portage Lake, there are two instabilities. We focus
on the larger and more active of the two, laying farther up-glacier, between $200$ and $1100\,m$ from the 2021 terminus. The instability shows clear signs of deformation including tension cracks, a main scarp, a talus source area, and antiscarps (Higman et al., 2023). Rockfall activity has been observed at the site and surficial streams on the slide disappear into the subsurface (Higman et al., 2023). InSAR data suggest the landslide moved between 2016 and 2022 (Schaefer et al., 2024). The instability is located in Chugach flysch (Upper Cretaceous age) and the lithology is sedimentary, composed primarily of metagraywacke
and metasiltstone (Wilson et al., 2015). The instability has a volume of around $11$ to $35\,Mio.\,m^3$ ($5$ to $19\,Mio.\,m^3$ according to Schaefer et al. (2024); Tab. D2 and App. D).

## A3 Barry

Barry Glacier is a tidewater glacier flowing southwest into Barry Arm in Prince William Sound, around $50\,km$ northeast of Whittier, Alaska (Fig. 1A and Fig. 2). The glacier flows from an accumulation area in a large ice complex in the Chugach
Mountains at $2700\,m\,a.s.l.$ to sea level. The region around the glacier terminus is very dynamic. There are a number of instabilities to the northwest and southeast of the terminus. We focus on the largest instability, which is characterized by anti- and normal-scarps, a main scarp, and a talus slope area (Higman et al., 2023). Multiple studies have confirmed the movement of this landslide since 2000 (Dai et al., 2020; Schaefer et al., 2024). Like Portage, the Barry instability is located in Chugach flysch. As of 2021, around half of the landslide toe was buttressed by the glacier, though the glacier has been retreating past the

instability since 2010. The Barry instability is the second largest in this study, with an estimated volume of 188 to 500 Mio. m$^3$ (117 to 564 Mio. m$^3$ according to Schaefer et al. (2024); Tab. D2 and App. D).

## A4  Yale

Yale Glacier is a tidewater glacier terminating in College Fiord in Prince William Sound, approximately 70 km northwest of Valdez, Alaska (Fig. 1A and Fig. 2). It flows from a large glacier complex in the Chugach Mountains at around 3600 m a.s.l. to
sea level. Within 2 km of the present-day terminus, there are three instabilities. We focus on the largest of the three, which has a volume between 255 and 750 Mio. m$^3$ (145 to 1025 Mio. m$^3$ according to Schaefer et al. (2024); Tab. D2 and App. D). The instability is characterized by anti- and normal-scarps, a talus slope area, shear zones, and tension cracks (Higman et al., 2023), and movement of the landslide has been confirmed with InSAR (Schaefer et al., 2024). As with the other nearby instabilities, the lithology of this site is Chugach flysch. The landslide started to become ice-free around 1977, but it took until 2021 for the
terminus to completely retreat past the toe area.

## A5  Columbia

Columbia Glacier, located around 35 km northwest of Valdez, Alaska (Fig. 1A and Fig. 2), is likely one of the most well-known glaciers worldwide due to its striking retreat over the past decades. The highest reaches of the glacier are nearly 3700 m a.s.l. in the Chugach Mountains and it flows to the south down to sea level in Prince William Sound. Near the present-day terminus of
Columbia Glacier, there are several instabilities to the north. We select the instability which is closest to the terminus (between 500 and 1800 m in 2021) and which could pose the largest hazard in the coming decades. The instability is characterized by tension cracks, normal scarps, a talus slope, and a lake which drains periodically (Higman et al., 2023). InSAR results from Schaefer et al. (2024) suggest the landslide moved between 2016-2022. The lithology of the instability is Chugach flysch. It has a volume of approximately 44 to 150 Mio. m$^3$ (17 to 111 Mio. m$^3$ according to Schaefer et al. (2024); Tab. D2 and App.
D).

## A6  Tyndall

Tyndall Glacier is located in the St. Elias mountain range and terminates at sea level in Taan Fiord, around 110 km northwest of Cordova, Alaska (Fig. 1C and Fig. 2). The glacier is bordered by some of Alaska's tallest mountains, with an accumulation area reaching up to 5290 m a.s.l.. There are a number of instabilities near the terminus of Tyndall Glacier. We select the one
that failed catastrophically in 2015, which had a volume of 63 Mio. m$^3$ and caused a tsunami with 193 m runup (Higman et al., 2018). The remaining sliding mass is around 100 Mio. m$^3$ (Tab. D2 and App. D). This site is thus distinct from the others because it has already experienced a catastrophic failure. The instability at Tyndall is located in the Kulthieth Formation, a sedimentary lithology from the Eocene composed of conglomerate-mudstone (Wilson et al., 2015). As of 2021, the glacier buttresses the length of the landslide toe. After a rapid retreat in the late 1980s, the glacier terminus was up-valley with respect
to the landslide but has been stable or slowly advancing since the early 1990s.

## A7 Alsek

Alsek Glacier is located in southeastern Alaska, around $100\,\mathrm{km}$ from Yakutat (Fig. 1D and Fig. 2). Alsek Glacier flows from an elevation of $2420\,\mathrm{m\,a.s.l.}$ down to $60\,\mathrm{m\,a.s.l.}$. The glacier terminates in Alsek Lake, which has grown from a few square kilometers in the 1950s to about $75\,\mathrm{km}^2$ today as the glacier has retreated (Loso et al., 2021). Between 3.5 and $5\,\mathrm{km}$ up-glacier from the 2021 terminus, an instability with a volume between 19 and $50\,\mathrm{Mio.\,m}^3$ (Tab. D2 and App. D) is found on the eastern side of a nunatak. The failure is characterized by a large head scarp which extends around $1\,\mathrm{km}$ laterally. The instability at Alsek Glacier is found in volcanic rocks of the Chugach accretionary complex, a metamorphic lithology from the Upper Cretaceous (Wilson et al., 2015). Gneiss, migmatite, and schist are the primary rock types (Wilson et al., 2015).

## A8 Grand Plateau

Grand Plateau Glacier is located in southeast Alaska, around $120\,\mathrm{km}$ from Yakutat (Fig. 1D and Fig. 2). It is a very large glacier, spreading over an area of $237\,\mathrm{km}^2$ and spanning an elevation range of nearly $4600\,\mathrm{m}$. The glacier currently terminates in three different lakes. We focus on the southeastern branch, where an instability with a volume of 50 to $150\,\mathrm{Mio.\,m}^3$ is present (Tab. D2 and App. D). The instability at Grand Plateau Glacier is located on an east-facing mountainside at the glacier left. The terminus began retreating past the instability around 2010, and the instability was completely debuttressed in 2022. It displays signs of movement through scarps, and like Alsek, it is located in volcanic rocks of Chugach accretionary complex.

## Appendix B: DEM years

**Table B1.** Summary of the years for which a DEM is available at the sites of interest. "BER" and "DEH" refer to the DEMs of Berthier et al. (2010) and Dehecq et al. (2020), respectively. Note that for the Dehecq et al. (2020) DEMs, multiple elevation models within a year may have been merged to generate the final product.

| Name | Year (BER) | Year (DEH) | Notes (BER) |
|---|---|---|---|
| Ellsworth | 1950 | - | |
| Portage | 1950 | 1980 | |
| Barry | 1957 | 1979-80 | |
| Yale | 1957 | 1979-80 | |
| Columbia | 1957 | 1979 | |
| Tyndall | 1972 | 1977 | very small portions in the accumulation area are from 1974 or 1976 |
| Alsek | 1948 | 1977-78 | |
| Grand Plateau | 1948 | 1977-78 | a portion of the accumulation area is from 1987 |

## Appendix C: Feature definition

### Instability-adjacent-glacier polygon

We created an instability-adjacent-glacier polygon to examine the glacier changes near the instability. For sites where the instability is far from the current terminus, we created a glacier polygon that is the same width as the instability. For sites where the instability is currently near the terminus, we manually created a polygon around $1\,\mathrm{km}$ long starting from the smallest extent and then intersected this with the glacier outlines from RGI Consortium (2017). Note that the smallest extent of the glacier is not necessary at present-day.

### Cross section

We generated cross sections through the landslide and glacier to study the glacier elevation changes near the instability. A line was drawn between the centroid of the instabilility polygon and the closest point on the centerline. Using the QGIS function "Extend lines," the line was extended by $4000\,\mathrm{m}$ in each direction. Points were then generated on the cross section at $10\,\mathrm{m}$ intervals using the QGIS function "Points along geometry." For these points, we then extracted values for each of the four available DEMs. For the cross section, we considered the area between the ridgetop behind the instability and the thickest part of the glacier.

## Appendix D: Landslide volume

In order to determine landslide volume, it was first necessary to determine the landslide extent. We delineated a polygon from high-resolution imagery or lidar by selecting all topographic features that showed signs of deformation. Subglacial and submarine extents were estimated using bedrock topography determined from the datasets mentioned in Section 3.4.1. Then, landslide volume was determined in two different ways: empirically and by expert estimation. We used an empirical relationship between the landslide surface area $A_S$ and its volume $V$ of the form

$$V = k \cdot A_S^{\beta}, \tag{D1}$$

where $k$ and $\beta$ are two empirical coefficients. We used three different sets of coefficients corresponding to a worldwide average for landslides, a worldwide average excluding shallow and submarine landslides, and an average for large landslides (Tab. D1). In addition, we completed expert estimation using the instability polygon and by inferring a sliding surface. To do so, the angle of the head scarp was extrapolated into the subsurface where available, and a concave failure plane was assumed in each case. We then estimated the volume between the basal failure plane and the surface.

**Table D1.** Different $\beta$ and $k$ coefficients used for estimating the volume of the instabilities $V$ based on the instabilities' surface area $A_S$ (Eqn. D1). "Number" is a key for the method in Table D2.

| Number | Source | $\beta$ | $k$ | Note |
|---|---|---|---|---|
| 1 | Guzzetti et al. (2009) | 1.450 | 0.074 | worldwide study |
| 2 | Jaboyedoff et al. (2020) | 1.362 | 0.288 | similar to (1) without shallow or submarine landslides |
| 3 | Jaboyedoff et al. (2020) | 1.375 | 0.410 | mean of range for large landslides |

**Table D2.** Geological characteristics of each site. "$V_{ls,X}$" are the estimated volumes of the instability, where X is either E for the volume determined using expert opinion, 1-3 using the empirical methods from Table D1, or S referring to the average of the volumes given by Schaefer et al. (2024). For Tyndall, the value in parentheses is the volume remaining at present-day. "$A_{ls}$" is the area of the instability from the delineated outline.

| | Ellsworth | Portage | Barry | Yale | Columbia | Tyndall | Alsek | Grand Plateau |
|---|---|---|---|---|---|---|---|---|
| $V_{ls,E}$ [$Mm^3$] | 150 | 35 | 500 | 750 | 150 | 160 (100) | 50 | 150 |
| $V_{ls,1}$ [$Mm^3$] | 66 | 11 | 188 | 255 | 44 | 62 | 19 | 50 |
| $V_{ls,2}$ [$Mm^3$] | 74 | 14 | 196 | 262 | 50 | 69 | 23 | 57 |
| $V_{ls,3}$ [$Mm^3$] | 127 | 24 | 339 | 454 | 86 | 118 | 39 | 98 |
| $V_{ls,S}$ [$Mm^3$] | 14-113 | 5-19 | 117-564 | 145-1025 | 17-111 | - | - | - |
| $A_{ls}$ [$km^2$] | 1.5 | 0.4 | 3.1 | 3.8 | 1.1 | 1.4 | 0.6 | 1.2 |

## Appendix E: Landslide activation

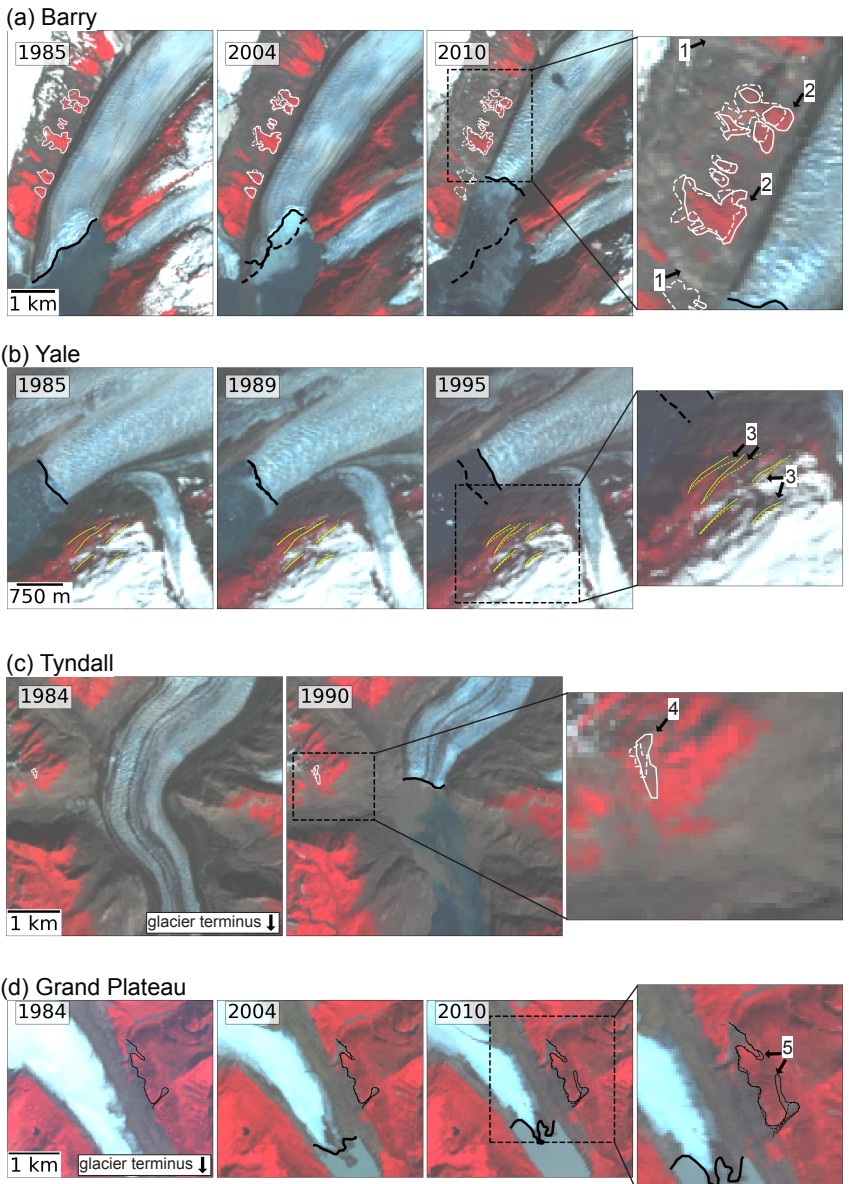

**Figure E1.** Landslide features and glacier terminus position during movement onset for sites where the terminus has passed the landslide. The year of acquisition of the individual satellite images is given at upper left. The panel at far right shows a zoomed version of the previous panel (shown by the black dashed box). Solid lines refer to the year at upper left in each image and for subsequent years, dashed outlines refer to the first time period (image at far left). Arrows highlight the formation of a head scarp (1), downward displacement of vegetation patches (2), downward displacement of surface features (3), widening of a crack (4), and the formation of a crack (5). Panels A-D refer to Barry, Yale, Tyndall, and Grand Plateau, respectively. Background images are Landsat 4-7 courtesy of the U.S. Geological Survey.

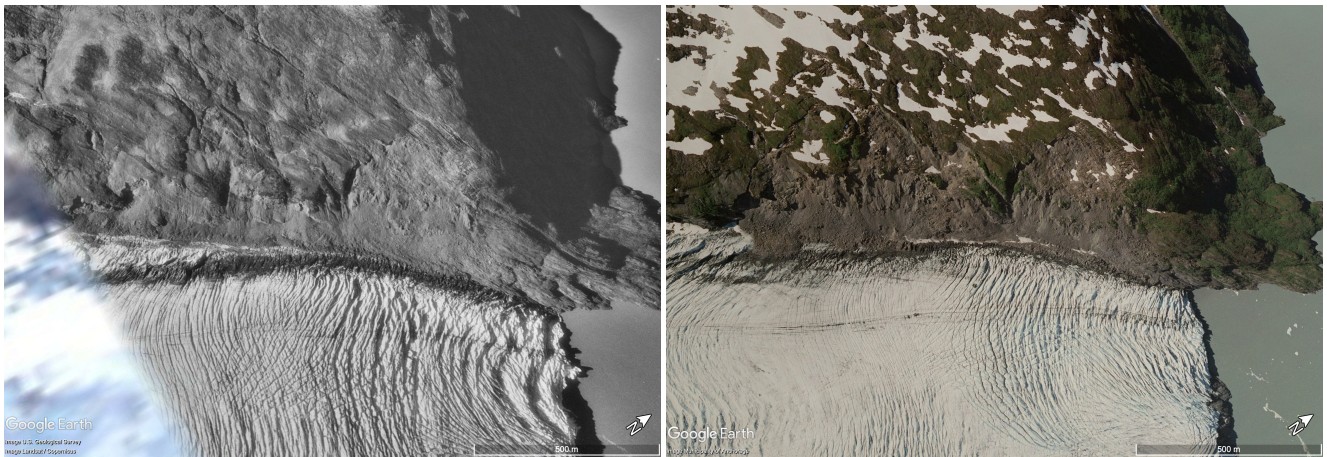

**Figure E2.** Images of Portage Glacier landslides in 1996 (left) and 2006 (right) show the evolution of the landslide throughout time and the disintegration of the slope. A scale bar and north arrow are at lower right. Both images are from Google Earth (Google Earth Pro, a, b).

## Appendix F:  Glacier characteristics

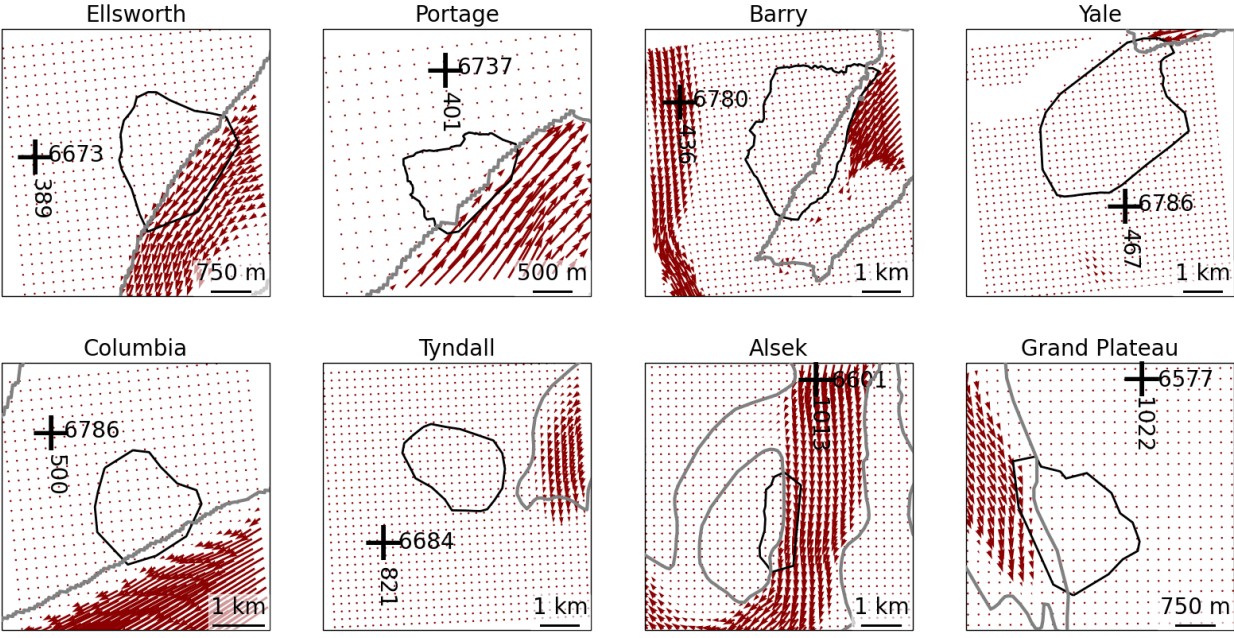

**Figure F1.** Average (1984-2022) displacement vectors from ITS-LIVE data. Glaciers are marked by gray (outlines from RGI Consortium (2017)) and black outlines mark the instability polygon. Coordinate crosses are given in UTM 6.

**Table F1.** Glaciological characteristics of each site. "$\Delta H_{gl,X}$" is the median elevation change of the glacier within the instability-adjacent glacier polygon (see App. C and Fig. 2 for definition), where X is either 1 for the period 1960-1978, 2 for 1978-2000, or 3 for 2000-2020. Note that i) the values for $\Delta H_{gl,1-2}$ for Ellsworth and Columbia refer to the period 1960-2000, and ii) the values for $\Delta H_{gl,3}$ are an average of the 2000-2005, 2005-2010, 2010-2015, and 2015-2020 changes. "$H_{gl,F}$" and "$H_{gl,M}$" are the median thicknesses of the glacier within the instability-adjacent glacier polygon from Farinotti et al. (2019) and Millan et al. (2022), respectively.

| | Ellsworth | Portage | Barry | Yale | Columbia | Tyndall | Alsek | Grand Plateau |
|---|---|---|---|---|---|---|---|---|
| $\Delta H_{gl,1}$ [$\mathbf{m\,a}^{-1}$] | | -2.7 | 1.25 | -5.61 | | -3.29 | -1.33 | -1.97 |
| | -1.22 | | | | -5.2 | | | |
| $\Delta H_{gl,2}$ [$\mathbf{m\,a}^{-1}$] | | -1.7 | 0.48 | -5.42 | | -9.55 | -1.81 | -3.3 |
| $\Delta H_{gl,3}$ [$\mathbf{m\,a}^{-1}$] | -2.51 | -1.62 | -9.65 | -2.18 | -14.1 | 0.56 | -2.85 | -5.85 |
| $\mathbf{H}_{gl,F}$ [m] | 220 | 170 | 230 | 210 | 660 | 120 | 260 | 480 |
| $\mathbf{H}_{gl,M}$ [m] | 250 | 200 | 280 | 260 | 400 | 240 | 280 | 260 |

**Appendix G: Meteorological analyses**

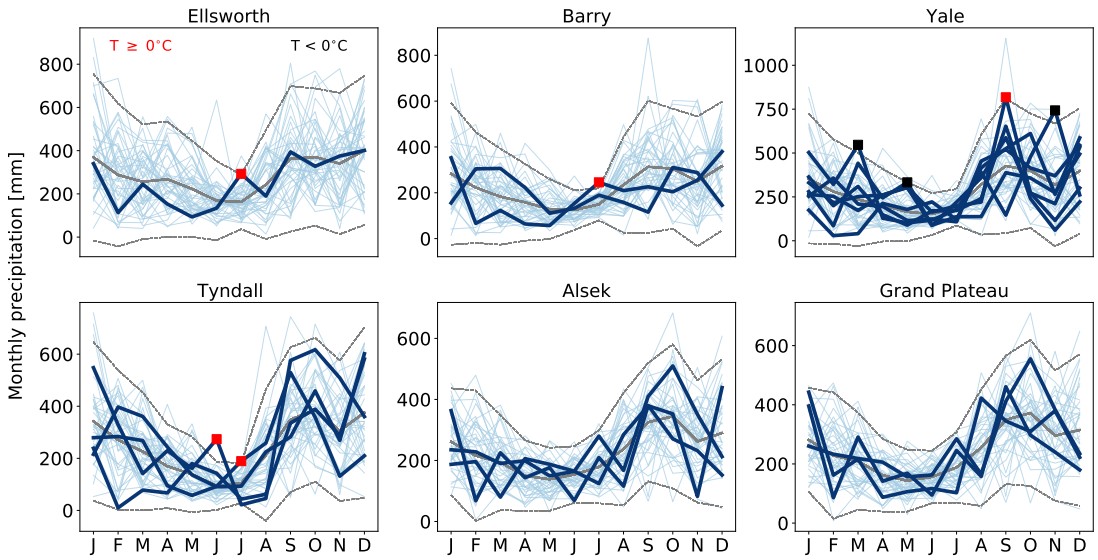

**Figure G1.** Average monthly precipitation at all sites which experienced a landslide acceleration during the study period. Thicker, darker blue lines correspond to the year or years during which the landslide accelerated, while light blue lines are all other years. The solid gray line corresponds to the mean monthly precipitation over the whole time period (1979-2022), and the gray dashed lines are two standard deviations above and below the mean. Red and black squares indicate temperatures above or below zero degrees, respectively, at instances where the monthly total during the activation period exceeded two standard deviations above the mean. Note the differing y-axes of the various subplots. All x-axes refer to the labels on the bottom row.

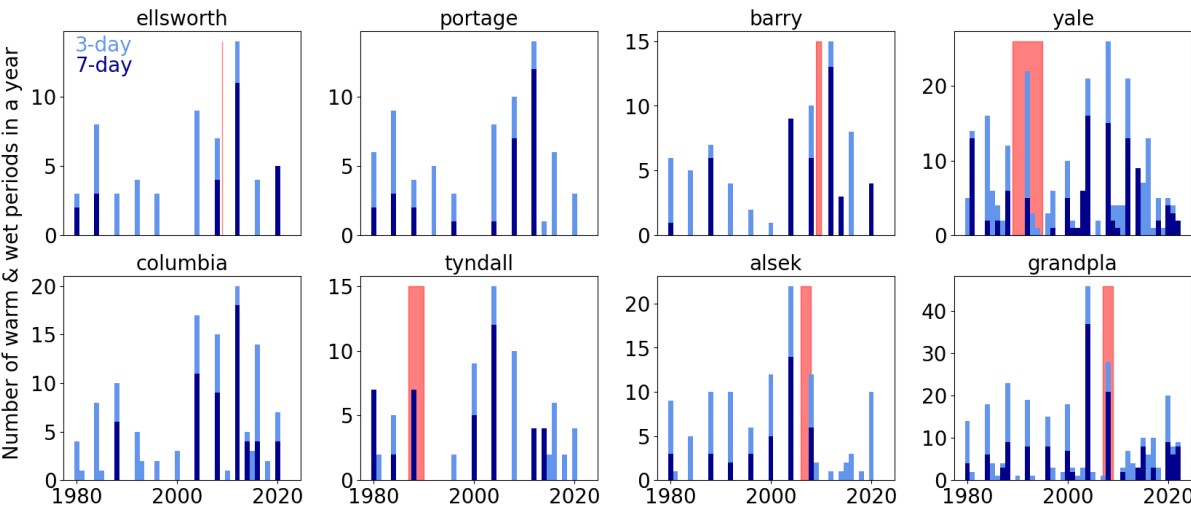

**Figure G2.** Occurrence of warm and wet periods in a year at the sites of interest. Light blue bars indicate the total number of days where the 3-day running precipitation total is greater than 3x the 90[th] precipitation quantile *and* the 3-day running temperature average was greater than $0°$C. Dark blue bars are the total number of days where the 7-day running precipitation total is greater than 7x the 90[th] precipitation quantile *and* the 7-day running temperature average was greater than $0°$C. Meteorological data comes from ERA5-Land (Muñoz-Sabater et al., 2021). In all panels, light red shading indicates the onset of landslide movement.

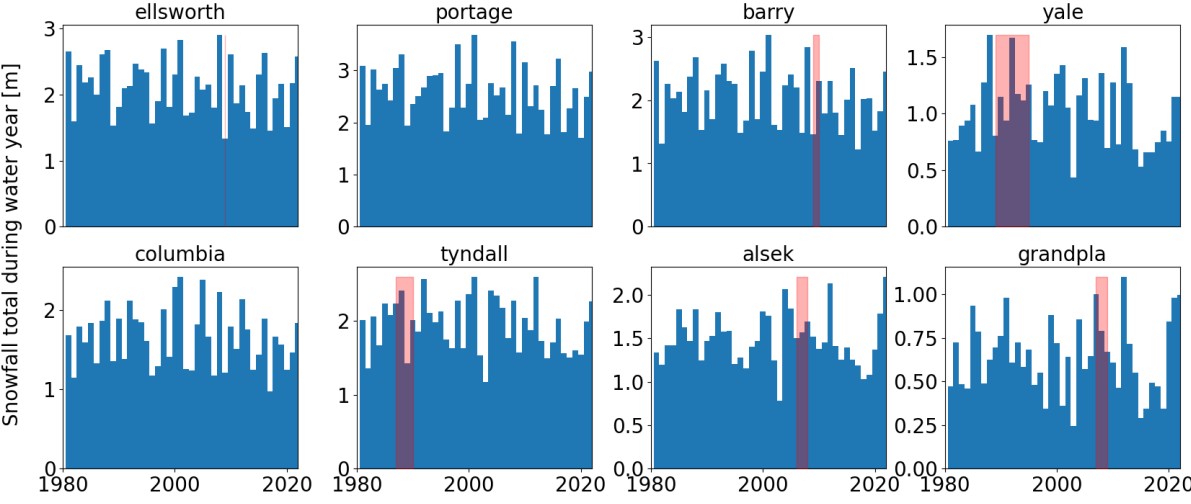

**Figure G3.** Snowfall total during the water year (Oct 1-Sep 30) at the sites of interest. Snowfall was calculated as the sum of all precipitation on days where the temperature was below $0°$C (data from ERA5-Land; Muñoz-Sabater et al. (2021)). In all panels, light red shading indicates the onset of landslide movement.

## Appendix H: Seismic analysis

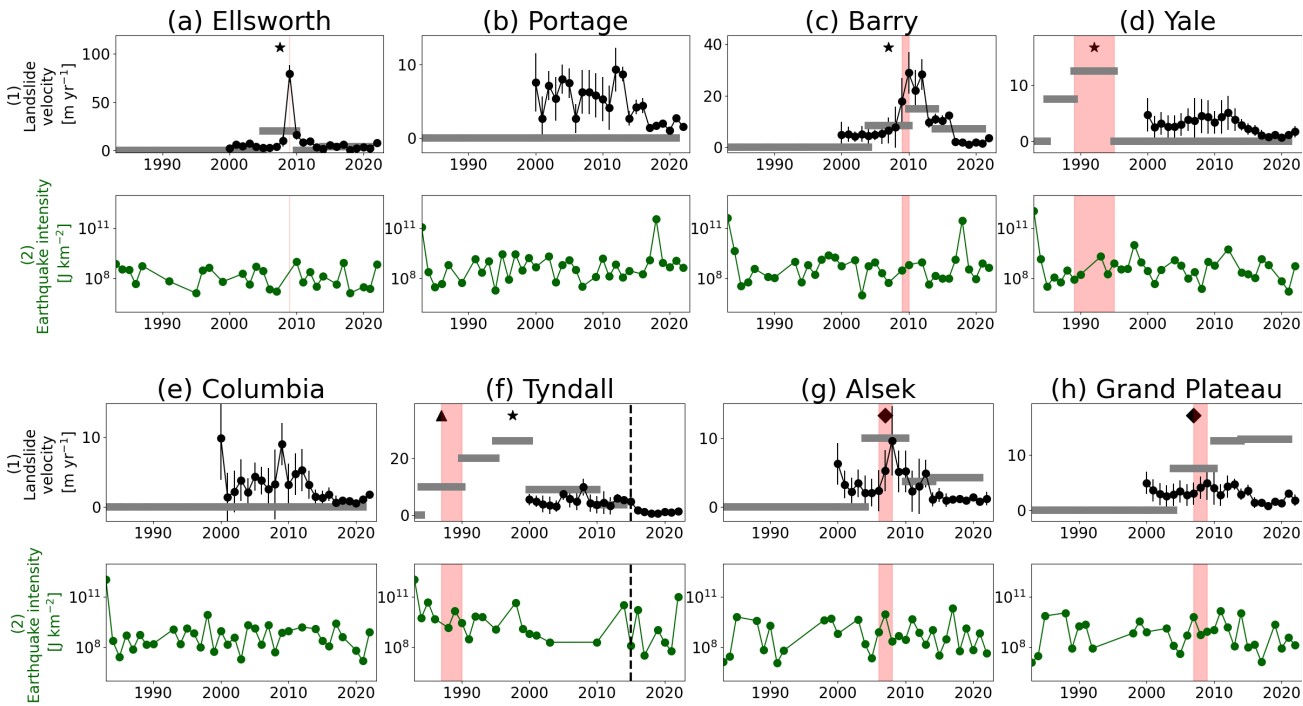

**Figure H1.** Landslide and earthquake intensity at the sites of interest. Row 1: Landslide velocities from ITS-LIVE (black circles, with vertical lines showing the uncertainty estimate from ITS-LIVE) and manual feature tracking (gray bars). Stars indicate onset of slope-wide deformation, triangles stand for crack opening, and diamonds mean both deformation and crack opening (in all cases the x-coordinate is the average over the time period in which the change was observed). Row 2: earthquake intensity at the instability (dark green), as calculated from U.S. Geological Survey (2023). In all panels, light red shading indicates the onset of landslide movement. At Tyndall, the black dashed line indicates a catastrophic failure. Note the differing scales on the y-axis for the individual sites.

*Author contributions.* MJ and BH conceptualized the study. JW completed the manual feature tracking analysis, mapped terminus locations, extracted ITS-LIVE velocities, and wrote the manuscript with the help of MJ and DF. BH provided landslide polygons and volumes, as well as helpful discussion about possible mechanisms. RH generated all co-registered DEMs. AM aided with the feature tracking and explanations of possible landslide mechanisms. DF supervised the overall work progress and helped in designing the figures. All co-authors read and provided feedback on the paper.

*Competing interests.* The authors declare that they have no conflict of interest.

*Acknowledgements.* We thank Maximillian Van Wyk de Vries and Amaury Dehecq for the helpful discussions about feature tracking, as well as Marit van Tiel for the useful input about glacier hydrology. We also thank Etienne Berthier and Amaury Dehecq for providing the 1960s and 1980s DEMs, respectively. Additionally, we thank Daniel Ben-Yehoshua for the interesting discussions about landslide-glacier interactions. BH acknowledges funding from the US National Science Foundation, award number 205210. RH acknowledges funding from the Swiss National Science Foundation, grant number 184634, as well as funding from NASA, award number 80NSSC22K1094. We would also like to thank Stuart Dunning, an anonymous reviewer, and the editor Andreas Günther for their helpful feedback which improved the manuscript.

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
