# Peer review of "Landslide activation during deglaciation in a fjord-dominated landscape: observations from southern Alaska (1984-2022)"

_EGUsphere, 2024_

## Author Comment (AC1)

**Response to Reviewer 1**

We would like to thank reviewer 1 for taking the time to read and provide feedback on this manuscript. Here, we respond to their comments point-by-point. Note that line numbers refer to the revised manuscript (unless otherwise noted) and quotations from the updated text are italicized and in quotations.

| # | Comment | Line # / Section | Reply |
|---|---------|------------------|-------|
| 1-1 | The main short-coming of this study concerns it's overarching premise, which is stated as a regional analysis. The concept of regional analysis is not clearly defined in terms of scientific goals or questions being addressed at that scale. The eight landslide objects selected represent 1% of the total landslide inventory (perhaps this is a little higher when only lake/fjord terminating glaciers in southern Alaska are considered), and hence the authors need to consider whether the outcomes of the study have any relevance to the regional-scale understanding of glacier retreat and landslide instability phenomena in the lakes and Fjords of southern Alaska. In regional landslide studies, the analysis of multiple influencing factors are usually used to estimate susceptibility or plausibility of a landslide occurrence, particularly in terms of identifying patterns and key driving factors, in either a spatial and/or temporal context. The study here lacks this type of broad rationality, but rather focuses on eight landslide case-studies each with their own unique conclusions. | Overall | Thanks for this feedback. We agree that casting this work as a 'regional' study may have been somewhat misleading, and we appreciate the suggestion to present this work instead as a case-study (comment 1-2). In reality, this work falls somewhere between a regional analysis and a case study. We have too few sites for a regional study and the level of detail with a case study is typically higher than what we do here. Nonetheless, we are able to compare and contrast the evolution of eight sites, as well as open up some questions for further research. We investigate multiple influencing factors (precipitation, seismic energy, glacier retreat versus glacier thinning, and geology) and while we don't look at susceptibility or predict where future landslides may form, we are able to draw some general conclusions from the eight studied sites. Specifically, we find that landslides bordering water following deglaciation may be at particular risk for acceleration. Further work to see if this holds for more landslides over a wider area would be interesting, but is outside the scope of this work. Concretely, we've changed the title, removed references to a 'regional' work, and have instead framed the paper as a comparison of eight interesting landslides with sometimes similar and sometimes differing conclusions. |
| 1-2 | Revise the scope and the purpose of the study to reflect the case-study oriented approach using the eight landslides selected. However, bear in mind that the main outcome(s) lack scientific strength, e.g. it is already well established in the scientific literature that glacier volume reduction (e.g. thinning/retreat) can lead to slope movements, and on rare occasions, catastrophic failure. | Overall | Thanks for the suggestion. See the response to comment 1-1. We have reframed the paper and removed any claims of this being a 'regional overview'. However, we respectfully disagree with the reviewer that the results lack scientific strength - there have been only few studies comparing annual landslide evolution with glacier changes in Alaska. Additionally, the reviewer says that the link between glacier volume reduction and slope movement is well established, but the disagreement in the scientific community can be evidenced by the comments of reviewer 2 of this paper, who said "[Debuttressing] is not universally accepted as a causal mechanism of slope failure." McColl et al., 2010 is the most well-known example questioning whether debuttressing can cause slope instability. The authors note that ice is ductile under low strain rates and thus |

| | | | cannot be a rigid buttress for a deforming rock mass. Some papers have cited and built upon this idea (McColl et al, 2013; Storni et al, 2020; Lacroix et al, 2022), while others have found clear evidence for linkages between glacier downwasting and landslide activity (Kos et al, 2016; Glueer et al, 2020; Lacroix et al, 2022). |
|---|---|---|---|
| 1-3 | The potential strength of this study lays in the elucidation of the regional distribution of the landslide/glacier interactions, and the broader (spatial) understanding of dominant driving factor. This should include a regional consideration of rainfall and seismic activity (landslide-fault proximity). The current study already goes in that direction but is limited. To highlight a point stated in the conclusions, there is a potentially broader use of ITS-LIVE data on a world-wide scale, but to demonstrate this potential the authors should focus on a truly regional study across the fjords of southern Alaska, for which there is an extensive landslide and glacier inventory. | Overall | Thanks for the comment. A comprehensive regional comparison of landslide occurrence and the potential influencing factors (precipitation, seismicity) is beyond the scope of this paper, which focuses on eight sites specifically. Future work, which is mentioned in the conclusion, will focus on landslide detection over a larger area using ITS-LIVE, as well as comparison with various factors (elevation, aspect, spatial distribution, precipitation, etc.) to determine correlations with landslide occurrence. To make this clearer, we have added the following sentence to the conclusion (lines 570-571):

"*Such a regional overview would allow for correlating landslide activity with various factors over a broader area, specifically elevation, aspect, precipitation amount, and proximity to faults, to name a few.*" |
| 1-4 | The manuscript is highly descriptive and long and could do with shortening/condensing, including moving non-essential information to a supplementary section. | Overall | Thanks for this suggestion. We have done a detailed read-through of the manuscript to eliminate superfluous information (e.g. repetition in the methods and limitations sections, ideas about tidal and wave influences, the changing ice to water interface, repetition in the results and discussion). We moved the detailed study site descriptions to the appendix as suggested in comment 1-14. We also moved Fig. 6 to the appendix, eliminated Fig. 7 by merging the precipitation information into Fig. 4 (and put the new supplemental plots in the appendix), and moved the updated Fig. 8 to the appendix. |
| 1-5 | The abstract is vague and requires more context with respect to the conclusions. As an example "..17 times higher compared to five years preceding the acceleration" (see line 10) – what is this referring to? The abstract needs to be more understandable. | Abstract | Thanks for this point. This was similar to feedback given by reviewer 2 (comment 2-2). We have rewritten the abstract and described the conclusions, and specifically the rates, in a more straightforward way (lines 8-15):

"*We found that the majority of landslides underwent a pulse of accelerated motion during the studied time period. In four cases, landslide movement coincided with the rapid retreat of a lake- or marine-terminating glacier past the instability. At these sites and during these accelerations, the glacier retreat rates were up to 7 times higher than average, while the landslides reached velocities that were up to 9 times higher than their long-term average. Two sites showed no movement, though both landslides are known to be moving at velocities below the detection threshold of the methods employed here. At two other sites where the landslides are still in* |

| | | | *contact with the ice, above-average precipitation and increased glacier thinning were found to coincide with accelerated motion, though conclusive causal links could not be drawn and the effect of short-term precipitation could not be ruled out."* |
|---|---|---|---|
| 1-6 | The discussion of the Tungnakvislarjökull landslide should be moved to the section dealing with landslides in an Alpine setting. | 50 | We've moved this sentence to the portion of the paragraph about alpine glaciers. |
| 1-7 | discussions on landslides in Alpine setting section, the relationship of glacier unloading and landslide movement was established by Kos et al (2016), the statement pertaining to altered groundwater creating critical conditions is a speculative assertion in Glueer et al (2020). | 50 | We've rephrased the sentence and added the Kos et al. (2016) citation. We also adjusted the wording of the groundwater statement to clarify that it's speculative. The resulting sentences read as follows (lines 62-66):

"*Studies of the Moosfluh landslide in Switzerland showed that landslide deformation can be related to debuttressing, with landslides reacting rapidly to glacier changes upon crossing a threshold of ice loss (Kos et al., 2016). Others found that the glacier controls the landslide velocity but has little effect on its stability (Storni et al., 2020), and (Glueer et al, 2020) suggested that altered groundwater conditions may lead to enhanced slope instability.*" |
| 1-8 | The potential landslide velocity is primarily controlled by litho-structural characteristics, buttressing ice is an external resisting factor with minor influence. | 55 | We are not totally sure to understand the reviewer's comment here. We believe the reviewer takes issue with the fact that the paragraph (lines 48-58 in the original manuscript) talks exclusively about debuttressing without referencing other factors (such as litho-structural characteristics) which are surely also relevant. To remedy this, we've rearranged the paragraph slightly to make the wording clearer. Additionally, we've added the following sentences to describe how debuttressing is still a debated topic (lines 45-48):

*"In fact, there has been some debate about whether glacier debuttressing can cause slope failure due to the viscous nature of ice at low strain rates (McColl et al., 2010; McColl and Davies, 2013; Storni et al., 2020). Others suggest that debuttressing can increase shear stress and act in combination with other processes such as rainfall to promote slope movement (Le Roux et al., 2009)."*

To highlight the importance of litho-structural characteristics, we've added the following to the same paragraph (lines 54-57):

"*In addition, litho-structural characteristics (Kuhn et al., 2023; Stead and Wolter, 2015), rock mass properties (Wang et al., 2021; Gischig et al., 2016; Hugentobler et al., 2022), and changing lake water levels (Hendron and Patton, 1987; Wang et al., 2008) are among the mechanisms which drive landslide motion. All of these* |

| | | | *processes—as well as combinations of them—may be relevant to the sites studied here*." |
|---|---|---|---|
| 1-9 | and Kos et al (2016) suggested that landslides react rapidly to glacier changes upon crossing a threshold of ice loss". | 55 | We are not sure to understand the reviewer's comment here, but we believe it's related to the fact that there were two citations within one sentence, and they were suggesting to make the latter into an in-text citation. We've reworked these sentences to reflect that change and they now read as follows (lines 62-66):

"*Studies of the Moosfluh landslide in Switzerland showed that landslide deformation can be related to debuttressing, with landslides reacting rapidly to glacier changes upon crossing a threshold of ice loss (Kos et al., 2016). Others found that the glacier controls the landslide velocity but has little effect on its stability (Storni et al., 2020), and (Glueer et al, 2020) suggested that altered groundwater conditions may lead to enhanced slope instability.*" |
| 1-10 | Is it correct to write thinning or retreat? Retreat and thinning occurs simultaneously, are there glaciers that retreat without thinning or vice versa? | 60 | Thanks for this point. While glaciers typically thin and retreat simultaneously, they can also thin while advancing (in the case of a surging glacier, for example). We refer to thinning primarily as vertical change in the glacierized area, and retreat as horizontal change. However, this is something which we discuss later on in the discussion (Section 5.2) and see that it's confusing in the introduction. We've therefore changed the wording to remove "retreat or thinning" and instead say "ice loss." |
| 1-11 | I have doubts whether this study is a *detailed regional* | 70 | We've adjusted the wording to remove references to a regional study and reframed the paper to be a case study (please see our response to comments 1-1 and 1-2). Here, we rephrased the sentence as follows (lines 79-81):

*"By doing so, we provide the first study comparing detailed glacier evolution---including both thinning and terminus retreat---with landslide movement in southern Alaska."* |
| 1-12 | "…and discuss in the context of the possible physical mechanisms behind the slope instabilities (sect 5)". What of the key factors geology and rock mass characteristics determining physical mechanisms? These are not featured nor discussed in the manuscript. | 75 | Rock mass characteristics are certainly relevant for slope stability. However, the remote-sensing-based approach that we use here does not allow for such a detailed consideration of the site-specific geology. Dealing with these factors is thus outside of the scope of this paper. I've added a sentence to the third paragraph of the introduction drawing attention to these relevant aspects (lines 54-57):

"*In addition, litho-structural characteristics (Kuhn et al., 2023; Stead and Wolter, 2015), rock mass properties (Wang et al., 2021; Gischig et al., 2016; Hugentobler et al., 2022), and changing lake water levels (Hendron and Patton, 1987; Wang et al., 2008) are among the mechanisms which drive landslide motion. All of these* |

| | | | |
|---|---|---|---|
| | | | *processes—as well as combinations of them—may be relevant to the sites studied here*." |
| 1-13 | why are the criteria selected important or relevant? There could certainly be more criteria to consider geological susceptibility, permafrost thermal state etc. These are very important factors (spatially) to consider in a regional study. | 95 | Based on this feedback, as well as feedback from reviewer 2 (comment 2-4), we've restructured this section to have less reliance on strict selection criteria. We are interested in large landslides since they can have larger inundation zones (Iverson et al., 1998, Griswold & Iverson, 2008, Chae et al., 2017), ones which border glaciers so that we can study the effect of the changing glacier conditions, and ones which have showed recent signs of activity and thus may pose a higher risk of acceleration or collapse.

We agree with the reviewer that geological susceptibility could be important too. However, this is very difficult to quantify and outside the scope of this work. Nonetheless, to address this point, we've added a sentence to the Section "Study Area" stating that all landslides are in sedimentary or metamorphic rock (lines 106-107):

*"All study sites are large landslides in sedimentary or metamorphic rock."*

We also pick up this point in the discussion (lines 444-445):

*"[…] three out of four sites are in sedimentary lithologies (Yale, Tyndall, and Barry), which are particularly susceptible to water intrusion due to high porosity (Selley, 2005)."*

Line 80 in the original manuscript draft stated that permafrost is unlikely, however we've now added an additional clause after the en dash to quantify this using data from Obu et al., 2018 (lines 89-91):

"*The large precipitation amounts result in a thick winter snowpack which, combined with relatively mild temperatures, make extended permafrost coverage unlikely – less than a 1% probability of occurrence at our sites according to Obu et al. (2018) (Fig. 1).*" |
| 1-14 | these descriptions could be moved to a supplementary section and/or tabulated so they are more easily read. | Section 2.1-2.8 | Thanks for the idea. In order to also shorten the manuscript as suggested (comment 1-4), we moved the descriptions to the Appendix and made a short, general description in the Study Area section. |
| 1-15 | legend for the geological base map should simply indicate the rock types (formation names are not particularly useful for those readers who don't know the local geology) | Figure 1a | Thanks for the suggestion. We changed the figure to have a more generalized form of the formation names, but did not change to rock type. From the 10 updated legend entries, over half contain information about the rock type (sedimentary & volcanic rocks, granitic rocks, etc.). One exception is "Chugach accretionary complex," a widespread unit in southern Alaska, and the lithology |

| | | | |
|---|---|---|---|
| | | | of this unit is described in the site description. Additionally, from the examples of other geological studies that we've seen, we noted that the formation name is typically given, not the rock type. We thus decided to stick with the formation name, but in a more generalized form as compared to the original manuscript. |
| 1-16 | Figure 1e is hard to read, perhaps make it into a separate larger figure. It would be useful to see the distribution of the landslide inventory more clearly | Figure 1e | Thanks for pointing this out. We've done some work on the color scheme to improve the readability of this figure, also based on feedback from reviewer 2 (comment 2-35). We've decided to keep panel e as a part of Figure 1, but we've added an enlarged version to the appendix so readers can see the landslide distribution more clearly. |
| 1-17 | Arrange the figure 4 sub components in the same order as figures 2 and 3. | Figure 4 | Changed as suggested. |
| 1-18 | How would a precipitation (long-term) trend be an important factor for landslide activation/failure? What is the relationship? E.g. these questions need to be placed in the context of the enormous rainfall that parts of southern Alaska receive. | 400 | Thanks for this comment. Indeed, we acknowledge that analyzing annual precipitation may have some limitations (see "Meteorology" in the Methods, as well as "Landslide interactions with the glacier and environment" in the Discussion). There, we say that landslides can be caused by both short-term, intense precipitation and long-term precipitation causing saturation of the subsurface. In order to be more thorough, we've additionally analyzed ERA5-Land daily data. We've added a new plot to the appendix (Fig. G2) which looks at warm and wet periods. However, no link was found between the daily data and landslide movement, primarily due to uncertainty in the landslide activation time. Additionally, we've decided to analyze annual precipitation anomalies in Figure 4. This allows for comparison of a particular year against the long-term average (1980-2009) and thus puts a single year into a longer-term context. |
| 1-19 | This figure is difficult to read, they could be much larger in size, and maybe show only one Landsat image as a reference, with glacier/landslide outlines for each year where there are changes observed. | Figure 6 | Per the suggestion of reviewer 2 (comment 2-40), we've decided to move this figure to the Appendix. We did increase the image size to the maximum possible. However, we chose not to use a single Landsat image because the multiple images show the evolution of various features (e.g. the development of a scarp or the appearance of a crack), and it is precisely this evolution which we aim to show. As the figure is now in the appendix, we decided to leave it as-is. |
| 1-20 | What is the significance of showing cumulative monthly precipitation plots in terms of landslide activation/failure when the dark blue lines correspond to other years (light blue lines) where no activation is recorded? Why isn't rainfall a significant conditioning/triggering factor in southern Alaska? | Figure 7 | We agree with the reviewer that this figure was difficult to interpret. Because of the new analyses performed in reply to reviewer's comment 1-18, we decided to remove this figure from the revised manuscript. |
| 1-21 | an untested hypothesis is merely speculation. Is there precedence in the | 415 | We agree that this statement was speculative. After reworking this section, removing Fig. 7, and including |

| | literature for this assertion? Either demonstrates the plausibility of a snow loaded slope triggering movement or leave it out. | | the precipitation time series in Fig. 4, we removed all references to snow load being a relevant mechanism for slope triggering. |
|---|---|---|---|
| 1-22 | triggering of a landslide would be associated temporally with the earthquake occurrence, but this relationship unfortunately cannot be shown. Lack of evidence in this case doesn't mean that seismic activity is not important, it may actually be more important than glacial retreat for landslide activation/ongoing failure – this should be expanded in the discussion. | Section 4.4.2 | Thanks for this comment. We had tried to address this in Section 5.2, where we conclude the discussion of seismic activity by saying the following (lines 485-486): *"While we do not find a direct link between seismic activity and slope accelerations in our data, we recognize that seismic events can contribute to the development of instabilities through a preconditioning of the related slopes."* To make this clearer, we've added the following sentence to Section 4.3.2 to direct the reader to the corresponding discussion section (lines 362-363): *"While we acknowledge that seismic shaking can cause rock damage which impacts landslide stability (see Sect. 5.2), the evidence here shows no direct link between specific seismic events and landslide acceleration."* |
| 1-23 | How reliable is it to compare ice thinning thresholds between Alpine landslide/glaciers and landslide/Fjord glaciers? How relevant is 100m of thinning at Moosfluh to the cases in southern Alaska. | 495 | Thanks for this good point. We added an additional sentence to Section 5.2 to account for the distinction between alpine and maritime glaciers, as well as land- versus water-terminating ones (lines 458-459): *"However, these numbers cannot be directly compared to the Moosfluh case, which is an alpine glacier in very different climatic conditions, but may imply region-specific thinning thresholds."* We've also added a conceptual figure (Fig. 6) which deals with the land- versus water-terminating cases and discuss the differences in Section 5.1. |
| 1-24 | On this point, please always refer to the primary literature where phenomena/relationships are first reported and then the later studies that find confirmation – the Glueer et al article falls in the latter. | 495 | The citation for Glueer was left as-is since the authors defined the threshold at 100m, but we've added a Kos et al. (2016) citation to the statement about landslide activation following glacier thinning to a critical thickness. The sentences now read (lines 453-456): *"Kos et al. (2016) proposed that landslide activation may begin after the glacier thins to a critical thickness. For the Moosfluh landslide in Switzerland, for example, Storni et al. (2020) found that slope displacements were larger where ice thickness was below 50 m and smaller where the ice was ca. 100 m thick, while Glueer et al. (2019) found that the whole landslide accelerated after the ice thinned below 100 m."* |
| 1-25 | The several factors that the authors did not consider are central to the discussion, and therefore need to be included in a regional study. | 560 | The factors that the reviewer is referring to are: site-specific structural geology, slope hydrology, and snow height. As mentioned in the text (line 564 in the original manuscript), the structural geology is not something |

| | | | that can be determined from remote sensing data and is therefore outside of the scope of this paper. Additionally, slope hydrology is closely related to the fracturing of the subsurface and thus also cannot be inferred from remote sensing. To address the impact of snow accumulation on slope stability, we have added a plot looking at annual snow totals (derived from meteorological data) to the Appendix (Fig. G3), as well as the following sentence in the text (lines 492-495):

*"Annual snowpack releases large amounts of water into the slope, which may impact stability. Detailed analyses on snow hydrology are outside the scope of this work, but an analysis of total solid precipitation over the time period 1980–2022 did not show a correlation with landslide activation (Fig. G3 in App. G)."* |
| --- | --- | --- | --- |

**References**

Chae, Byung-Gon, Hyuck-Jin Park, Filippo Catani, Alessandro Simoni, and Matteo Berti. "Landslide Prediction, Monitoring and Early Warning: A Concise Review of State-of-the-Art." Geosciences Journal 21, no. 6 (December 2017): 1033–70. https://doi.org/10.1007/s12303-017-0034-4.

Glueer, Franziska, Simon Loew, and Andrea Manconi. "Paraglacial History and Structure of the Moosfluh Landslide (1850–2016), Switzerland." Geomorphology 355 (2020): 106677. https://doi.org/10.1016/j.geomorph.2019.02.021.

Griswold, J.P., and Iverson, R.M., 2008, Mobility statistics and automated hazard mapping for debris flows and rock avalanches (ver. 1.1, April 2014): U.S. Geological Survey Scientific Investigations Report 2007-5276, 59 p.

Iverson, Richard M., Steven P. Schilling, and James W. Vallance. "Objective Delineation of Lahar-Inundation Hazard Zones." Geological Society of America Bulletin 110, no. 8 (August 1998): 972–84. https://doi.org/10.1130/0016-7606(1998)110<0972:ODOLIH>2.3.CO;2.

Kos, Andrew, Florian Amann, Tazio Strozzi, Reynald Delaloye, Jonas Ruette, and Sarah Springman. "Contemporary Glacier Retreat Triggers a Rapid Landslide Response, Great Aletsch Glacier, Switzerland." Geophysical Research Letters 43, no. 24 (December 28, 2016). https://doi.org/10.1002/2016GL071708.

Lacroix, Pascal, Joaquin M. C. Belart, Etienne Berthier, Þorsteinn Sæmundsson, and Kristín Jónsdóttir. "Mechanisms of Landslide Destabilization Induced by Glacier-Retreat on Tungnakvíslarjökull Area, Iceland." Geophysical Research Letters 49, no. 14 (July 28, 2022). https://doi.org/10.1029/2022GL098302.

McColl ST, Davies TRH, McSaveney MJ. 2010. Glacier retreat and rock-slope stability: debunking debuttressing. Geologically active : delegate papers 11th Congress of the International Association for Engineering Geology and the Environment, Auckland, Aotearoa, 5-10 September 2010. Auckland, New Zealand; pp. 467-474.

McColl, Samuel T., and Timothy R. H. Davies. "Large Ice-Contact Slope Movements: Glacial Buttressing, Deformation and Erosion: SLOPE MOVEMENT; GLACIER DEFORMATION, EROSION AND ENTRAINMENT." Earth Surface Processes and Landforms 38, no. 10 (August 2013): 1102–15. https://doi.org/10.1002/esp.3346.

Storni, Enea, Marc Hugentobler, Andrea Manconi, and Simon Loew. "Monitoring and Analysis of Active Rockslide-Glacier Interactions (Moosfluh, Switzerland)." Geomorphology 371 (December 2020): 107414. https://doi.org/10.1016/j.geomorph.2020.107414.

---

## Author Comment (AC2)

**Response to Reviewer 2**

We would like to thank reviewer 2 for taking the time to read and provide feedback on this manuscript. Here, we respond to their comments point-by-point. Note that line numbers refer to the revised manuscript (unless otherwise noted) and quotations from the updated text are italicized and in quotations.

| # | Comment | Line # / Section | Reply |
|---|---------|------------------|-------|
| 2-1 | This is a really interesting study that adds to our understanding of how large slopes behave in very transient parts of the landscape, that, despite their activity remain poorly quantified. Please take the following as constructive, and, as a means to stimulate discussion which can only improve the final uptake of this good piece of work and drive future work to test your ideas. | Overall | Thanks a lot for this kind comment, and for the constructive feedback on our work! We appreciate the time you took to have a detailed read through our paper and provide thorough feedback. |
| 2-2 | I do believe the abstract could be far tighter and would benefit from a rewrite, especially when talking about rates – try to keep this to the take home messages. | Abstract | We've taken the reviewer's suggestion to rewrite the abstract. Specifically, we simplified the part about the rates, which now states the following (lines 10-11):

 *"At these sites and during these accelerations, the glacier retreat rates were up to 7 times higher than average, while the landslides reached velocities that were up to 9 times higher than their long-term average."* |
| 2-3 | Although the case-study landslides are from a 'region', I do not think that can be considered a 'regional' analyses. Eight is not enough to unpick a regional pattern/differences or the behaviour of lake, marine or still ice contact slopes. | Overall | We agree, the portrayal of this study as 'regional' was, in hindsight, somewhat misleading. This was also a concern raised by reviewer 1 (see comment 1-1). Based on the suggestions from both reviewers, we have reframed the paper to be a case study rather than a regional overview, and have removed all references to the latter (including changing the title). We are convinced that the shift in focus resulted in a much-improved manuscript. |
| 2-4 | I would like some further justification for the selection of the 8, how are they representative of the 780, rather than some (rare?) end member? For example, at Ellsworth you note a number of landslides, so, what is the rationale to only characterise one, rather than a suite of them to see if the behaviour (and lags to ice front change) are comparable or if you are picking earlier phases/time zero of instability further up ice? Can you say why a minimum of 10 million volume was used? Landslides far smaller are able to cause damaging landslide-tsunami unless you can say otherwise? | Study Site | This is similar to a comment from reviewer 1 (1-13) and prompted us to reformulate the third paragraph of the 'Study Area' section significantly. We are careful not to suggest that the eight study sites are representative of the rest of the landslide inventory. We did this by putting less focus on the inventory itself and also by acknowledging that the site selection was *"relatively arbitrary, focusing on sites that stood out as worth investigating from early versions of the inventory"* (line 102).

 In the detailed site descriptions (now App. A), we do describe why we chose a particular landslide at sites where there are multiple landslides from which to choose. Rather than focus on multiple instabilities at |

| | | | one site, we sought to compare several landslides in various glaciological, hydrological, meteorological, and seismological situations throughout southern Alaska. Finally, we removed references to a volume threshold, instead saying that all sites are 'large.' While small landslides can also be damaging, we focus on large landslides, which can have larger inundation zones (Iverson et al., 1998, Griswold & Iverson, 2008, Chae et al., 2017) and could displace more water when impacting lakes or fjords. |
|---|---|---|---|
| 2-5 | Something that needs to be dealt without throughout is the behaviour of large rockslopes, and, the science of creep that may or may not transition to an eventual catastrophic failure – this would give more context to the role of speeds ups, ie, are they significant, or, not suggestive of any move towards (or away from) some eventual failure that can cause the cascading risks you note. | Overall | Thanks for the suggestion. It's true that we present evidence of slope acceleration without providing much context about what that means for the risk of catastrophic failure. We have added a few sentences about creep to the discussion (lines 501-507): *"While slow movement may continue for a long period of time, landslides can experience periods of acceleration and some even progress to catastrophic failure (Lacroix et al., 2020), with Tyndall being an example of the latter. The reason why some landslides maintain a slow velocity and others speed up or fail is unknown, but slow motion is typically observed prior to failure (Hendron and Patton, 1987; Handwerger et al., 2019b; Federico et al., 2011). This transition from slow to fast movement may be related to decreasing porosity in the shear-zone (Agliardi et al., 2020; Iverson et al., 2000), decreasing viscosity of the landslide material (Mainsant et al., 2012; Carrière et al., 2018), or shear localization (Voight, 1988; Lacroix and Amitrano, 2013) and is a topic of ongoing research."* We also note how the evolution following initial movement is varied (lines 497-498): *"However, the initial landslide response to does not determine the long-term evolution of the landslide, where both a re-stabilization or catastrophic failure might occur."* |
| 2-6 | The authors use 'debuttressing' very rapidly in the introduction without critique. It is not universally accepted as a causal mechanism of slope failure (particularly in bedrock) but at various points you seem to be ascribing the word as a process – rather than saying a slope became ice-free for example in line 141. You return to this later, but, I'm not convinced 'debuttressing' can be applied, rather than the more correct statement later that ice-loss / thinning has been associated with data and modelling to instability and changes in landslide motion. | 141 | Ballantyne 2002 defined debuttressing as the "removal of the support of adjacent glacier ice during periods of downwastage". We therefore understand this to be the process through which slopes become progressively less supported during glacier mass loss. However, there is quite some debate about this. As mentioned in our response to reviewer 1 (comment 1-2): "McColl et al., 2010 is the most well-known example questioning whether debuttressing can cause slope instability. The authors note that ice is ductile under low strain rates and thus cannot be a rigid buttress for a deforming rock mass. Some papers have cited and built upon |

| | | | this idea (McColl et al, 2013; Storni et al, 2020; Lacroix et al, 2022), while others have found clear evidence for linkages between glacier downwasting and landslide activity (Kos et al, 2016; Glueer et al, 2020; Lacroix et al, 2022)." |
|---|---|---|---|
| | | | Nonetheless, the way it was written in the original manuscript was confusing so we adjusted the wording in the introduction as follows (lines 24-26): |
| | | | *"Glacier retreat, which removes support from adjacent valley walls in a process termed ``glacier debuttressing'' (Ballantyne, 2002), may lead to the destabilization or failure of weakened valley slopes."* |
| | | | Additionally, we have changed the statement from line 141 (original manuscript) to now read (lines 614-615): |
| | | | *"The landslide started to become ice-free around 1977..."* |
| 2-7 | Similarly, 'paraglacial' landslide formation is used in line 70 with no setup as to what you mean by the use of this term or original references to the term. Are you implying that these landslides did not exist prior to the onset of (this) deglaciation? | 70 | We did not intend to imply that these landslides were initiated during this deglaciation, acknowledging that the landslides may have already existed for a long time (lines 423-426):

 *"It is possible that the underlying structures of these landslides have existed for many decades, centuries, or even millennia, and previous work has shown that landslides can reactivate or even fail millennia after deglaciation (Hermanns et al., 2017). We thus do not speak about landslide initiation, which would imply the motion onset, nor do we rule out that earlier phases of activity may have existed."*

 Nonetheless, we agree that it was confusing to use paraglacial without a definition. We defined paraglacial using the definition of Church & Ryder 1972 (lines 72-73):

 *"(We use "paraglacial" to define non-glacial processes impacted by glaciation (Church and Ryder, 1972))."*

 This simply implies a linkage to glacier changes but not on a specific time frame. |
| 2-8 | I like the reuse of ITS-LIVE for a new purpose, but, are yearly data too crude to characterise the questions – can you provide a little justification for the use of annual data, and the limitations of doing so? You say in Line 204 that using a larger timestep may show more (perhaps part of | Methods | We argue that annual velocity data are sufficient because we are interested in interannual to decadal time changes rather than daily, monthly, or seasonal changes. The usage of annual data may result in some smoothing of velocity peaks, but allows us to evaluate the evolution over a longer time period. |

| | | | |
|---|---|---|---|
| | the rationale for annual data as the finest step), but I disagree automatic feature tracking would not cope with this, is it a matter of the image temporal separation that you feed into automatic optical feature tracking. | | This was poorly explained in the original manuscript. This reasoning is now reflected at lines 140-143:

*"Since we focus on landslide evolution over several decades—going back to the start of the satellite era—we leverage yearly and multi-year velocities to characterize landslide changes. The usage of annual data may result in the loss or smoothing of the signal, especially during times of rapid movement, and it does not allow us to see seasonal effects. However, this temporal resolution allows for characterization of long-term trends and interannual changes."*

The statement about automatic feature tracking was poorly worded - we meant to say that slow-moving landslides may have small movement which is not detectable on annual time scales - so we decided to remove this sentence. |
| 2-9 | I am also surprised that the ice velocity product is not used as a supplementary dataset, if might prove to have little use and be ruled out, but, I'd expected to see velocity changes of the 'buttressing' ice alongside other plots – can you say why these data are not considered useful? It is clear from some work that ice-velocity can responds to landslide displacement into the glacier, and, it may (debate!) be that velocity of the ice may feedback to the velocity of the landslide(s). It would also be good to see data on Fig 3, is there a glacier response in the velocity field as the landslide deforms into/under the ice? | 204 & Figure 3 | Thanks for this point. Indeed, while slope deformation is impacted by glaciers, glaciers also respond to slope deformation. Prompted by this comment, we created a new figure where the velocity vectors over the glacier are visible (see Fig. F1 in the Appendix). We did not include this in Fig. 3 because the magnitude of the vectors varied so much that the two scales (landslide and glacier) were incompatible. In the new figure, we looked for areas where the 1984-2022 average velocity vectors might be displaced due to the landslide movement. We would argue that Barry shows a slight change in glacier flow direction in response to the landslide, and potentially Grand Plateau as well. We've commented on this in Section 4.2 (lines 309-310 & 314-315):

*"Deformation of the terminus in response to the landslide movement is clearly visible from the ITS-LIVE velocity vectors (Fig. F1 in App. F)."*

*"Like Barry, slight deformation of the terminus in response to the landslide is visible (Fig. F1 in App. F)."* |
| 2-10 | In the results, text and Figure 4 I can't reconcile the red bars with the velocity of the landslides, most are clearly moving before the red bar, sometimes for years, how have you decided that this red bar is the onset? | Figure 4 & Methods | The placement of the red bar was done manually and the timing represents the initial pulse of significant, slope-wide deformation during our study period. We did this by finding the earliest image pair over which the slope deformation increased dramatically. We have added a sentence to the methods to make this clearer (lines 152-153):

*"The ``activation period'' was defined manually to be the timing of an initial pulse of significant, slope-* |

| | | | *wide deformation during our study period (Sect. 5.2)."* |
|---|---|---|---|
| | | | We agree with the reviewer that in the previous version of the manuscript, the determination of this activation period was not obvious due to high uncertainty in the ITS-LIVE data prior to 2000. We have now decided to use ITS-LIVE velocities only after 2000 and manual feature tracking throughout the whole time period (see reply to comment 2-11), and hope that this clarifies the positioning of the red bars. |
| 2-11 | There seems to be considerable disparity in the ITS-LIVE and manual data, for example, Grand Plateau in Fig 4, which one you use fundamentally changes the interpretation. E..g for G Plateau it is either decreasing in velocity as the terminus retreats to the landslide, and then becomes quite constant or, it is near zero then starts to accelerate as the terminus reaches the landslide. There are similar things to resolve with many of the 8 (Tyndall is another where the data are showing two very different patterns, increasing to 2000 with the manual, or, decreasing to ~2005 from a peak. | Methods | We agree with the reviewer that the interpretation for the early part of our time series is dependent on the considered velocity dataset. Because the ITS-LIVE data is sparse and has high error prior to 2000, we decided to remove these data from the analysis, now relying exclusively on manual feature tracking prior to 2000 and both manual and automatic feature tracking after 2000. We have updated Figure 4 accordingly. Grand Plateau is one case where a disparity between the manual and automatic feature tracking remains. We suspect in this particular case that the feature tracking algorithm is not able to cope with the large changes in the slope. In the manual feature tracking, we observed some partial slope failures, as well as significant changes to the vegetation on the slope. The automatic feature tracking may have failed to identify these changes. We've added a sentence on this to Section 4.1 (lines 274-275): *"We suspect that this discrepancy is due to the slope appearance changing drastically and significant vertical motion, both of which pose challenges for the feature tracking algorithm."* |
| 2-12 | The precipitation analyses seems to be setting up a model that the landslide must be responding to the rainfall in that year? Why not be plotting some form of precipitation anomaly or metric on a figure like 4 that shows the two variables as a time series? I would want to look at landslide responses to rainfall over time, is there any, and, is it consistent or does any relationship break down after some 'key' event? | Methods | Thanks for the suggestion. We've added an additional row to Figure 4 where the annual precipitation anomaly with respect to the long-term average is plotted, and also includes information about the annual temperature. |
| 2-13 | I think the same is true of the seismic data presented, it is an interpreted derivative of velocity you are showing – the accelerated portion, rather than presenting these data to show the full time series links (or not links looking at it). If there is very little link, | Methods | Thanks for the comment. As suggested, we've made a time series of seismic energy which can be more easily compared to the slope evolution. Since no link to the displacement time series was found, we put this in the Appendix and wrote the following in Section 4.4.2 (lines 360-361): |

| | does it warrant being in the main text, or, in the SI with a few lines saying the seismic intensity time series showed little? | | *"We did not observe a temporal correlation between high seismic intensity and landslide velocity during the years of landslide activation, nor did we observe increased landslide activity in the years following a particular seismic event."* |
|---|---|---|---|
| 2-14 | For both precip and seismic I do wonder if the annual time series approach to deformation may mask any links. | Methods | Thanks for this feedback. We acknowledge that annual precipitation may not be the best metric to find links with landslide activation. The primary difficulty is uncertainty in the timing of the landslide activation which, in the worst case, was defined to be within a period of 6 years. Nonetheless, we tested several different metrics: annual precipitation anomalies with respect to the long-term average (Fig. 4), average monthly precipitation (Fig. G1), number of warm & wet periods in a year (Fig. G2), and total snowfall during the water year (Fig. G3). We did not find a clear link between landslide activation and precipitation at any scale (with the exception of Ellsworth as mentioned in the text). |
| 2-15 | Discussion: Really interesting ideas deserving of further investigation. | Discussion | Thanks for the kind feedback on this section! We agree that further studies, especially focusing on landslide activation near rapidly retreating water-terminating glaciers, would be valuable and relevant for hazard mitigation in the near future. |
| 2-16 | You talk of landslide activation, but, as a generalisation here, you are dealing with landslides that were already 'features' at the start of the analyses, rather than capturing the true activation/initiation and associated conditions? I wonder (and you should cover this) how representative these forms of landslide are compared to those that show no precursory creep of note (or do they and it has been missed) and fail catastrophically? | Discussion | The distinction between "initiation" and "activation" is presented at the start of Section 5.2. Here, we acknowledge that these features have existed for a long time and we are likely seeing "reactivation" of the landslide rather than its initiation. Based on Lacroix et al., 2020, it is common for slow motion to be observed prior to failure. For the case of landslides that do not show creep prior to failure, we would argue that this is due to missed observations rather than a lack of movement. As requested in another comment (2-5), we added the following sentences about creep to the discussion (lines 503-507):

 *"The reason why some landslides maintain a slow velocity and others speed up or fail is unknown, but slow motion is typically observed prior to failure (Hendron and Patton, 1987; Handwerger et al., 2019b; Federico et al., 2011). This transition from slow to fast movement may be related to decreasing porosity in the shear-zone (Agliardi et al., 2020; Iverson et al., 2000), decreasing viscosity of the landslide material (Mainsant et al., 2012; Carrière et al., 2018), or shear localization (Voight, 1988; Lacroix and Amitrano, 2013) and is a topic of ongoing research."* |
| 2-17 | I wonder around lines 525 onwards where you think about differing controls you | Discussion | Thanks for the suggestion. As proposed, we've added a conceptual figure at the start of the |

| | | | |
|---|---|---|---|
| | should be posing a conceptual diagram to bring these ideas together and allow you to pose questions/ideas around the future of these instabilities. | | discussion (Fig. 6). This figure shows 3 different states: a) glaciated conditions, b) during de-glaciation, and c) upon glacier disappearance. In each case, a hillslope is drawn. Prior to glacier retreat, the glacier and corresponding water table are shown, along with the saturation of the subsurface and the pressure on the failure surface. During glacier mass loss, the glacier surface lowers, as does the water table. After the glacier is gone, we show two cases: 1) a landslide in an empty valley where excess pore water pressure cannot develop under normal conditions and 2) a landslide bordering a body of water where the subsurface remains saturated following retreat. |
| 2-18 | The change from landslide-ice to landslide-water, you say that this changes toe saturation. Are these not wet/warm glaciers where the margins and base are likely to be very wet already? What is instantly (on a geological time scale) is the yield stress of what the slope is resting against as you say. Is that more or less likely to induce failure as compared to a land terminating removal of ice at the slide margins? | Discussion | Thanks for the feedback. We believe the conceptual figure mentioned in the previous comment (2-17), and the corresponding elaboration in the discussion, have addressed these questions. Namely, we examine how the subsurface hydrology and stresses would evolve between glaciated and non-glaciated conditions, as well as with and without proglacial water bodies. |
| 2-19 | Lines 583 around tidal/wave influences needs to be supported by your data. | 483 | We have removed this paragraph since it cannot be supported by our data and is somewhat speculative. This additionally addresses a comment from reviewer 1 about the manuscript being lengthy (1-4). |
| 2-20 | It feels a shame to end the discussion with a following section on the limitations, it is quite a negative finish, it would be nicer to keep this with the methods, as, that is when I had those questions to ask (unless you have to match journal convention). | Discussion | This section, which was originally in the Discussion, has now been dissolved and added to various subsections in "Data & Methods." The feature tracking limitations, for example, are incorporated into Section 3.1 "Landslide displacements", and the downside to using annual precipitation data is mentioned in Section 3.4.2 "Meteorology." In the Discussion, one paragraph discussing the limitations remains in Section 5.2. |
| 2-21 | Few cases studies in depth, are you saying there is a far larger population of landslide-tsunami that are unstudied, or, there have been few? I'm also unclear if you are ending the paragraph referring to all large landslides associated with a glacier, or, the ones that enter water bodies specifically. | 39 | Thanks for pointing out that this was unclear. We wanted to say that few studies have looked at landslide evolution over deep fjords, and even fewer have looked in combination with glacier evolution. We have rephrased the sentence as follows (lines 42-43): *"Despite the destructive potential of these events, few studies have investigated landslide evolution near deep fjords with a specific focus on the glacier evolution."* |
| 2-22 | I'm not sure I recognise post-failure instabilities. They are either an instability, or, a relict landslide/mass movement deposit? | 63 | Sorry for the confusing wording here. What we should have said instead of "instabilities ... post-failure" was "relict landslide" or "mass movement |

| | | | deposit." We've thus changed the sentence to read (lines 73-75):

*"The increased awareness of this hazard spurred the creation of an Alaskan landslide inventory (Higman, 2022; Higman et al., 2023), which documents instabilities, relict landslides, and mass movement deposits throughout the state."* |
|---|---|---|---|
| 2-23 | I presume it is not only retreat, but also thinning that is contributing this mass | 63 | Correct, we've changed "glacier retreat" to "glacier mass loss" (line 69). |
| 2-24 | Jumping between 'local' SE Alaska numbers of relevance e.g. uplift, to National (e.g. earthquakes). | 89 | We see how this was unclear. We've added uplift numbers for the Kenai Peninsula and adapted text to be less specific about southeastern Alaska. |
| 2-25 | You can say 'may' control, or, set this as a formal hypothesis to test, but as written it looks like you have concluded that thinning/retreat DO control 'mobility', a term I'm not sure is what is often used, over deformation, creep, motion etc, as the norm is often that mobility is around the final position of a failed mass? | 178 | Thanks for this good point about the terminology. We changed "...glacier thinning and retreat control the mobility..." to "...glacier thinning and retreat may control the motion...". |
| 2-26 | Why are they alternative drivers? We know precip and seismics can have an effect, so, why not here assume that all may play a role, but some might be more significant, and, there is the possibility that some may actually compound the deformation process-response? | 184 | We agree that the wording "alterative drivers" was imprecise, and that these factors may act together to contribute to deformation. We changed "alternative" to "co-" and "mobility" to "deformation" (line 117):

*"we evaluated time series of precipitation and seismicity as possible co-drivers of landslide deformation."* |
| 2-27 | I presume ITS-LIVE is optimised for the glacier areas rather than the boundaries of scene pairs where edge effects may play a part, do the data extend far beyond the slide boundaries, with equal errors to the rest? | 186 | Thanks for bringing up this point. Data do extend far beyond the slide and glacier - there's a buffer of ca. 25 km around all glaciated areas. It's also worth noting that we're dealing with the mosaic product and not the scene pairs directly.
We did a quick analysis to examine the error on and off glacier. At Ellsworth, we computed the ratio between the error and velocity over the landslide area and compared this to the corresponding ratio over a branch of less dynamic glacier ice at the site. We see that the relative error tends to be higher over non-glacierized areas, which we attribute to the lower velocities over these areas. The absolute error over the landslide and glacier are comparable. Since the landslides are moving at typically lower velocities as compared to the glacier so the relative error is higher. During the time that the landslide accelerated, we see that the relative error drops and is lower than the relative error over the glacier at the same time. |
| 2-28 | But, you might be looking for retrogressive behaviour or evidence of future upslope instability? | 209 | The reviewer's comment focuses on manual feature tracking over time periods where a catastrophic failure was observed. In our case, this is about Taan Fiord specifically. Our original formulation said, "For |

| | | | |
|---|---|---|---|
| | | | slopes that experienced catastrophic failure, no feature tracking was done **after** the failure because the fundamental changes of the sliding mass make the pre- and post-failure slopes incomparable." What we meant is that it is not possible to do feature tracking between images pre- and post-failure because the mass is fundamentally different and it is unclear what feature would be tracked and over what distance (for example, using the distance from the original mass to the glacier surface or the fjord). To make this clearer, we changed the aforementioned sentence as follows: "no feature tracking was done **during** the time period containing the failure". At Taan Fiord, specifically, we have a pre-failure image in 2014 and one post-failure in 2021 and thus did not do manual feature tracking over this time. |
| 2-29 | I'm not clear what you mean by the average of individual features, do you use a statistical treatment of the per-pixel displacements, or, a subset of these that have been quality controlled? Do you use off-landslide pixels as a control measure? | 211 | In answer to the first question: we intended to say that for each mapped feature, we take several measurements and then average those distances to get the displacement. We've rephrased this as follows (lines 155-157):

*"For each landslide feature indicating slope-wide movement (e.g. crack opening, displacement of vegetation patches), we took around three to five measurements of the distance moved between two images. The median of these measurements, divided by the number of years between the images, gives the slope speed for that period."*

In response to the second question: no, we do not use off-landslide pixels as a control. However, the selected images have good co-registration so we expect the off-landslide pixels to be stationary. |
| 2-30 | Happy to talk to you about break-point/change-point analyses to statistically begin to ID the rapid phases in a quant/reproducible way. | 239 | Thank you for this offer and suggestion. While we agree this would be more rigorous compared to our manual identification of the phases, the tests we conducted using the ruptures package in python were unsuccessful. In particular, the package failed to identify any rupture for six out of our eight sites. More specifically, we were able to find a set of parameters which detected the accelerations at Ellsworth and Barry. The acceleration at Alsek was quite short-lived, so this doesn't stand out particularly in the ITS-LIVE data and thus was not detected. At Grand Plateau, there is disagreement between the ITS-LIVE and manual feature tracking, so automatic detection would not work here. Additionally, two accelerations took place prior to 2000, and thus rely on the manual feature tracking data which has a much coarser temporal resolution and is thus not suitable for automatic detection. |

| 2-31 | Give indication of when in the year you make these measures without the need to look at SI please. | 248 | We added a sentence about the images coming primarily from July-October (lines 193-195):

*"The vast majority of the images used were taken between July and October, though occasionally a winter image was used if no summer image was available."* |
|---|---|---|---|
| 2-32 | There is still debate on the role of shaking: https://agupubs.onlinelibrary.wiley.com/doi/full/10.1029/2021JF006242 | 285 | Thanks for point this out. We've adapted the sentence to now read (lines 232-234):

*"We seek to quantify the effect of seismic activity on slope stability, since earthquakes may induce rock mass fatigue and promote failure (Gischig et al., 2016), though shaking can influence rock strength in a variety of ways, ranging from decreased to increased strength (Brain et al., 2021)."* |
| 2-33 | Not including the manual measurements? | 303 | We understand this comment to mean that the reviewer would like us to make a similar comparison for the manual feature tracking as we did for the ITS-LIVE velocities in the first paragraph of the 'Landslide evolution' section. We thus added the following sentence (lines 254-256):

*"During periods of acceleration, the landslide velocities from ITS-LIVE increased up to a factor of 9 compared to the average velocity between 2000 and 2022. Manual feature tracking, on the other hand, gave maximum velocities up to 7 times higher than the long-term average (1984-2022)."* |
| 2-34 | not convinced by pers. Comm. part given they are co-authors, can data snapshot not be down in the SI? | 581 | As suggested by the reviewer, we removed the previous 'personal communication' citation. We now directly reference the EarthScope/Unavco website with the Columbia GPS data, as well as show a snapshot in the Supplementary Information (Fig. 1). |
| 2-35 | On (e) having ice as white when the ocean also is does not work. Faults (and the difference in dashed lines) needs to be in the legend. The landslide locations A-D are as capitals, the panel letters are lower case. Move the p.frost prob lower, the legend does not sit well in the figure layout. (a) It might be my PDF version, but the water as grey is a bit odd and counter-intuitive, suggest some coherence with water as suggested in (e) and perhaps some version of blue. The geology legend has unequal line spacing and second line indent. I would also expand b, c, d so that the panels match the width of a. Is there more use of the ice on the panels to be had, perhaps use ice-velocity data? Thre are no coordinate crosses in b-d? | Figure 1 | Thanks for these helpful suggestions to improve this figure. In response to this comment, we: 1) changed the ocean to dark blue; 2) added faults to the legend (note there is no difference in the dashed lines - they all have the same line style - but some are overlapping and thus appear solid); 3) we changed the landslide locations to lowercase letters to match the panels; 4) moved the permafrost legend lower down in the plot and changed it to mean annual ground temperature because a divergent color scheme allowed for more contrast with the other colors; 5) changed the water color to dark blue in panels (a) and (e); 6) we left the line spacing as-is - it appears uneven because some legend entries are on two lines, we removed the second line indent; 7) we expanded panels (b), (c), and (d) and added coordinate crosses; 8) since there is already a lot of information contained in this plot, we decided against adding ice velocity since this would add |

| | | | another colorbar (though we've added it to Fig. 2, see answer to comment 2-36). |
|---|---|---|---|
| 2-36 | Ice velocity data semi transparent may be helpful? Personal preference perhaps, but I do not like the coordinate crosses at all (same in Fig 1), I would rather an outside grid/ticks and a legend encased in solid white. When are the Planet Labs images from? Summer 2023? I hope it becomes clear in the text how you determine the subglacial extent of instability as it is a really interesting open question. | Figure 2 | As suggested, we added ice velocity data semi-transparent in red. We decided to leave the coordinate crosses because we found that ticks or a grid on the border took up too much space with 8 different sites. The date of the Planet images is now written in the caption: "Satellite imagery from August 2023 over the eight sites". We hope that the reviewer is pleased with our description of the determination of the subglacial topography provided in Section 3.4.1 and the landslide extent in App. D. |
| 2-37 | It seems a wasted opportunity not to plot the glacier velocity product on, is there evidence of ice deformation in response to where the slide mass is? | Figure 3 | We decided not to plot the glacier and landslide velocity vectors together on this plot because the differing scales of the two mean that landslide velocity vectors would not be visible if glacier velocity vectors were plotted as well. For this reason, we've left Fig. 3 as-is but we've added a plot to the Appendix (Fig. F1) which shows some deformation of the glacier due to the landslide at Barry, and potentially Grand Plateau as well. We've drawn attention to this in Section 4.2 (lines 309-310 & 314-315):

*"Deformation of the terminus in response to the landslide movement is clearly visible from the ITS-LIVE velocity vectors (Fig. F1 in App. F)."*

*"Like Barry, slight deformation of the terminus in response to the landslide is visible (Fig. F1 in App. F)."* |
| 2-38 | A very nice figure. I do wonder on other ways to plot some of these data in addition…..perhaps cumulative displacement, do be more in keeping with retreat, and/or, the ice thickness data, again, cumulative rather than a rate – it depends what you think is the important driver, the absolute change in thickness relative to the landslide, or the rate of change? I would like to see 'major' events incorporated e.g. crack opening using symbols. | Figure 4 | Thanks for the suggestion. We decided to leave the displacement time series as-is since computing the cumulative displacement would cumulate the error as well. We would argue that both the absolute ice thickness change AND the rate of change are relevant for driving the landslide; we thus decided to add ice thickness as a second time series to one of the existing rows in Figure 4. We've also used symbols to indicate crack opening and slope-wide deformation. |
| 2-39 | I am not clear on how you decided the lower limit for the landslides? All the subglacial reconstructions suggest a steep trough ie no landslide toe. Is this likely 'real' ie they are scoured away, or, a function of the inversion process to yield thickness? | Figure 5 | The lower boundary of the landslide was inferred using subglacial topography from the datasets discussed in section 3.4.1 (Ice Thickness). Since this was not explicitly described in the manuscript, we've added the following sentence to Appendix D (lines 678-679):

*"Subglacial and submarine extents were estimated using bedrock topography determined from the datasets mentioned in Section 3.4.1."* |

| | | | The steep trough is likely a function of the inversion process rather than a real signal. |
|---|---|---|---|
| 2-40 | Sorry, this figure I struggle with and I would see moved to SI. | Figure 6 | We moved Fig. 6 to the Appendix as suggested. However, since it was not entirely clear to us what caused the struggle, we left the figure unchanged. |

**Sources**

Ballantyne, Colin K. "Paraglacial Geomorphology." Quaternary Science Reviews, 2002, 83.

Chae, Byung-Gon, Hyuck-Jin Park, Filippo Catani, Alessandro Simoni, and Matteo Berti. "Landslide Prediction, Monitoring and Early Warning: A Concise Review of State-of-the-Art." Geosciences Journal 21, no. 6 (December 2017): 1033–70. https://doi.org/10.1007/s12303-017-0034-4.

Church, Michael, and June M Ryder. "Paraglacial Sedimentation: A Consideration of Fluvial Processes Conditioned by Glaciation." GSA Bulletin 83, no. 10 (October 1, 1972): 3059–72. https://doi.org/10.1130/0016-7606(1972)83[3059:PSACOF]2.0.CO;2.

Glueer, Franziska, Simon Loew, and Andrea Manconi. "Paraglacial History and Structure of the Moosfluh Landslide (1850–2016), Switzerland." Geomorphology 355 (2020): 106677. https://doi.org/10.1016/j.geomorph.2019.02.021.

Griswold, J.P., and Iverson, R.M., 2008, Mobility statistics and automated hazard mapping for debris flows and rock avalanches (ver. 1.1, April 2014): U.S. Geological Survey Scientific Investigations Report 2007-5276, 59 p.

Iverson, Richard M., Steven P. Schilling, and James W. Vallance. "Objective Delineation of Lahar-Inundation Hazard Zones." Geological Society of America Bulletin 110, no. 8 (August 1998): 972–84. https://doi.org/10.1130/0016-7606(1998)110<0972:ODOLIH>2.3.CO;2.

Kos, Andrew, Florian Amann, Tazio Strozzi, Reynald Delaloye, Jonas Ruette, and Sarah Springman. "Contemporary Glacier Retreat Triggers a Rapid Landslide Response, Great Aletsch Glacier, Switzerland." Geophysical Research Letters 43, no. 24 (December 28, 2016). https://doi.org/10.1002/2016GL071708.

Lacroix, Pascal, Alexander L. Handwerger, and Grégory Bièvre. "Life and Death of Slow-Moving Landslides." Nature Reviews Earth & Environment 1, no. 8 (July 21, 2020): 404–19. https://doi.org/10.1038/s43017-020-0072-8.

Lacroix, Pascal, Joaquin M. C. Belart, Etienne Berthier, Þorsteinn Sæmundsson, and Kristín Jónsdóttir. "Mechanisms of Landslide Destabilization Induced by Glacier-Retreat on Tungnakvíslarjökull Area, Iceland." Geophysical Research Letters 49, no. 14 (July 28, 2022). https://doi.org/10.1029/2022GL098302.

McColl ST, Davies TRH, McSaveney MJ. 2010. Glacier retreat and rock-slope stability: debunking debuttressing. Geologically active : delegate papers 11th Congress of the International Association for Engineering Geology and the Environment, Auckland, Aotearoa, 5-10 September 2010. Auckland, New Zealand; pp. 467-474.

McColl, Samuel T., and Timothy R. H. Davies. "Large Ice-Contact Slope Movements: Glacial Buttressing, Deformation and Erosion: SLOPE MOVEMENT; GLACIER DEFORMATION, EROSION AND ENTRAINMENT." Earth Surface Processes and Landforms 38, no. 10 (August 2013): 1102–15. https://doi.org/10.1002/esp.3346.

Storni, Enea, Marc Hugentobler, Andrea Manconi, and Simon Loew. "Monitoring and Analysis of Active Rockslide-Glacier Interactions (Moosfluh, Switzerland)." Geomorphology 371 (December 2020): 107414. https://doi.org/10.1016/j.geomorph.2020.107414.